# Hybrid Belief–Reinforcement Learning for Efficient Coordinated Spatial Exploration

## Abstract

Coordinating multiple autonomous agents to explore and serve spatially heterogeneous demand requires jointly learning unknown spatial patterns and planning trajectories that maximize task performance. Pure model-based approaches provide structured uncertainty estimates but lack adaptive policy learning, while deep reinforcement learning often suffers from poor sample efficiency when spatial priors are absent. This paper presents a hybrid belief-reinforcement learning (HBRL) framework to address this gap. In the first phase, agents construct spatial beliefs using a Log-Gaussian Cox Process (LGCP) and execute information-driven trajectories guided by a Pathwise Mutual Information (PathMI) planner with multi-step lookahead. In the second phase, trajectory control is transferred to a Soft Actor-Critic (SAC) agent, warm-started through dual-channel knowledge transfer: i) belief state ~~initialization~~ transfer supplies spatial uncertainty, and ii) replay buffer seeding provides demonstration trajectories generated during LGCP exploration. A variance-normalized overlap penalty enables coordinated coverage through shared belief state, permitting cooperative sensing in high-uncertainty regions while discouraging redundant coverage in well-explored areas. The framework is evaluated on a multi-Unmanned Aerial Vehicle (UAV) wireless service provisioning task. Results show 10.8% higher cumulative reward and 38% faster convergence over baselines, with ablation studies confirming that dual-channel transfer outperforms either channel alone.

## 1 Introduction

Spatial exploration under uncertainty arises in numerous domains where autonomous agents must learn an unknown spatial field while simultaneously optimizing a task objective. Applications include environmental monitoring, dynamic wireless connectivity scaling, precision agriculture, disaster response, and infrastructure inspection (McEnroe et al., 2022; Popović et al., 2024). A fundamental challenge is that the spatial field of interest is heterogeneous and *a priori* unknown to the agents. Unlike settings where the spatial distribution is known or can be specified in advance, agents must simultaneously learn the underlying field while optimizing task performance. This gives rise to the coupled exploration-exploitation problem.

Using multiple agents offers advantages including expanded coverage, increased redundancy, and load balancing. However, co-ordination between mobile agents introduces significant challenges, including: (i) joint trajectory optimization over high-dimensional continuous action spaces; (ii) coordinating multiple agents through shared spatial awareness; (iii) balancing information acquisition against task performance; and (iv) sample-efficient learning without prior knowledge of spatial structure. These challenges are compounded by partial observability, where agents can only identify demand within their immediate sensing region (Popović et al., 2024).

Two dominant paradigms have emerged. Model-based approaches employ spatial statistical models to maintain probabilistic beliefs, enabling principled uncertainty quantification (Krause et al., 2008). However, these methods typically rely on myopic planning and do not adapt based on accumulated experience. Model-free deep reinforcement learning (DRL) can learn complex coordination policies, but suffers from poor sample efficiency (Wang et al., 2022). This dichotomy motivates our work: *combining the sample efficiency of*

*Bayesian spatial modeling with the adaptive policy learning of deep RL.* Our framework combines two standard constructs from the RL literature: the *belief state* (Kaelbling et al., 1998), a summary of the agent's uncertainty over the environment, and *learning from behavioural demonstrations*, which seeds policy learning with expert trajectories.

## 1.1 Contributions

We make the following contributions:

- We propose a hybrid framework combining Log-Gaussian Cox Process (LGCP) spatial modeling with Soft Actor-Critic (SAC) reinforcement learning for coordinated spatial exploration under unknown demand.

- We introduce a dual-channel warm-start mechanism for sample-efficient learning: (i) ~~belief initialization~~ belief state transfer, which provides the RL agent with an informed prior for early policy updates, and (ii) *behavioral transfer*, which seeds the replay buffer with LGCP-generate exploration trajectories. Ablation studies show that behavioral transfer provides the dominant benefit, while ~~belief initialization~~ belief state transfer yields additional gains when combined with replay buffer seeding.

- We employ uncertainty-driven Pathwise Mutual Information (PathMI) planning for non-myopic trajectory optimization during the exploration phase, extending standard informative path planning (IPP) with staleness-weighted revisitation incentives.

- We propose a variance-normalized overlap penalty that adapts coordination strength to local belief uncertainty, permitting cooperative sensing in high-uncertainty regions, while penalizing redundant coverage.

- We instantiate and evaluate the proposed framework using multi-UAV wireless service provisioning as an exemplar scenario, demonstrating up to 10.8% higher reward and 38% faster convergence versus baselines.[1]

The remainder of the paper is organised as follows. Section 2 reviews related work and, in Section 2.6, synthesises prior approaches to motivate the design choices underlying the proposed framework and situates the contribution within the broader literature. Section 3 formalises the system model, including the spatial demand field, the LGCP belief model, and the reward structure. Section 4 presents the proposed HBRL framework, detailing PathMI planning, the dual-channel knowledge transfer mechanism, and the SAC training procedure. Section 5 evaluates the framework through comprehensive ablation studies and comparisons with eight baselines. Section 6 concludes and discusses limitations and future directions.

## 2 Related Work and Motivation

### 2.1 Informative Path Planning

Informative path planning (IPP) addresses trajectory optimization for information acquisition under uncertainty. Krause & Guestrin (2007) propose an exploration–exploitation framework for single-agent sequential observation selection under unknown Gaussian Process (GP) kernel parameters. Relatedly, Krause et al. (2008) prove that mutual information for GP regression is submodular, yielding a greedy approximate algorithm for static multi-sensor placement. Both, however, are restricted to discrete candidate sets and do not extend to continuous action spaces for mobile robot path planning.

Cao et al. (2023) propose CAtNIPP, a learning-based framework for adaptive informative path planning that uses attention mechanisms to encode a global belief and guide non-myopic decisions, but in a single-agent setting only. Chen et al. (2024) address non-stationary GPs for robotic information gathering, again

---

[1]All code and research artifacts will be made openly available upon acceptance of the manuscript for publication.

considering single-agent scenarios only. Stephens et al. (2024) incorporate safety constraints into GP-based exploration, but do not address coordination among multiple agents. While these methods provide principled uncertainty-driven exploration, they predominantly focus on single-agent scenarios with pre-specified GP hyperparameters.

## 2.2 Spatial Statistical Modeling

For count data exhibiting spatial correlation, Log-Gaussian Cox Processes (LGCPs) model spatially varying intensity fields with Bayesian uncertainty quantification (Møller et al., 1998). Computational tractability is achieved through Gaussian Markov Random Field (GMRF) approximations (Rue & Held, 2005; Lindgren et al., 2011) and Integrated Nested Laplace Approximation (INLA) (Rue et al., 2009). Diggle et al. (2013) provides comprehensive treatment of LGCPs for spatial point patterns.

Recent advances include proposing fast maximum likelihood estimation using automatic differentiation (Dovers et al., 2023), and demonstrating LGCP fitting via GAM software (Dovers et al., 2024). LGCPs are widely used for spatial point process modeling, particularly in environmental and epidemiological applications (Jones-Todd & van Helsdingen, 2024; D'Angelo et al., 2022). Although, LGCPs provide principled uncertainty quantification for count data, their integration with robotic path planning or reinforcement learning remains relatively underexplored.

## 2.3 Reinforcement Learning-Based Coordination

Reinforcement learning (RL) has emerged as a promising paradigm for coordinated spatial exploration, with UAV coordination being a prominent application. The survey by Bai et al. (2023) identifies sample efficiency as a critical challenge. Wang et al. (2022) apply DRL for trajectory design in mobile edge computing (MEC) systems, demonstrating improved performance but reporting slow convergence.

Heterogeneous multi-agent reinforcement learning (MARL) has recently been proposed for joint trajectory and communication design but does not incorporate spatial priors (Zhou et al., 2024). Rizvi & Boyle (2025) develop shared Deep Q-Network (DQN) with action masking for Non-Orthogonal Multiple Access (NOMA)-UAV networks noting that fixing cluster sizes limits adaptability to varying user distributions. Zhao et al. (2025) employ graph-based MARL for search and tracking but focus on target localization rather than demand learning. Without structured spatial priors, these RL-based coordination methods suffer from poor sample efficiency in unknown environments.

## 2.4 Transfer Learning and Warm-Starting

Transfer learning accelerates RL by leveraging knowledge from related tasks or demonstrations. Hester et al. (2018) propose Deep Q-learning from Demonstrations, seeding replay buffers with expert trajectories. However, this requires access to expert demonstrations unavailable in novel environments. The survey by Zhu et al. (2023) identifies domain adaptation as a key challenge.

Skrynnik et al. (2021) address forgetful experience replay for hierarchical RL but focus on single-agent navigation. Wang et al. (2024) demonstrate warm-starting for hybrid vehicle energy management, showing significant sample efficiency gains. While these methods improve sample efficiency, the use of structured Bayesian spatial priors as the knowledge source for coordinated spatial exploration remains unexplored.

## 2.5 Trajectory Optimization

Trajectory design for mobile agents, particularly in UAV-enabled communications, has received substantial attention. Zeng et al. (2019) minimize propulsion energy for rotary-wing UAVs using successive convex approximation but assume perfect knowledge of user positions throughout the mission. Wu et al. (2018) jointly optimizes trajectory and power allocation; however, the approach requires full channel state information which may not always be available. Hao et al. (2024) address task offloading with trajectory control but do not incorporate online demand learning. Song et al. (2023) apply evolutionary multi-objective RL for UAV-assisted ~~mobile edge computing (MEC)~~ but focus on computational offloading rather than demand

learning. These methods assume known demand distributions, or employ full state knowledge, rendering them unsuitable when demand must be learned online from partial observations.

### 2.6 Research Gaps and Motivation

The reviewed literature reveals five key limitations that motivate our work:

- Classical trajectory optimization assumes known user distributions (Zeng et al., 2019; Wu et al., 2018; Hao et al., 2024). In many practical deployments, demand patterns are initially unknown and must be learned online.

- Many IPP methods employ greedy one-step planning (Krause & Guestrin, 2007). While computationally efficient, myopic strategies fail to anticipate future information gain, leading to suboptimal long-horizon performance.

- DRL approaches lack principled uncertainty estimates (Wang et al., 2022; Zhou et al., 2024; Rizvi & Boyle, 2025). Without calibrated uncertainty, agents cannot distinguish "unexplored" from "low-demand" regions.

- Pure DRL suffers from poor sample efficiency without spatial priors (Bai et al., 2023). Agents must discover demand structure through extensive trial and error.

- Much of the IPP and spatial modeling literature focuses on single-agent scenarios (Popović et al., 2024; Chen et al., 2024). Coordination among multiple agents requires redundant coverage avoidance and cooperative sensing strategies, which are not addressed by these methods.

**Table 1:** Summary of related work and key limitations. UD: Unknown Demand, MA: Multiple mobile Agents, NM: Non-Myopic Planning, SU: Spatial Uncertainty, SE: Sample Efficiency.

| Reference | Approach | UD | MA | NM | SU | SE |
|---|---|---|---|---|---|---|
| Krause et al. (2008) | Greedy Submodular placement | ✓ | ✗ | ✗ | ✓ | n/a |
| Wu et al. (2018) | Joint traj. & resource | ✗ | ✓ | ✓ | ✗ | n/a |
| Zeng et al. (2019) | Convex Opt. Min. flight energy | ✗ | ✗ | ✓ | ✗ | n/a |
| Wang et al. (2022) | DRL traj. for MEC | ✓ | ✓ | ✓ | ✗ | ✗ |
| Cao et al. (2023) | Learning-based IPP | ✓ | ✗ | ✓ | ✓ | ✓ |
| Zhou et al. (2024) | Heterogeneous MARL | ✓ | ✓ | ✓ | ✗ | ✗ |
| Hao et al. (2024) | Task offloading & traj. | ✗ | ✓ | ✓ | ✗ | ✗ |
| Rizvi & Boyle (2025) | SDQN + action masking | ✗ | ✓ | ✓ | ✗ | ✗ |
| **This Work** | HBRL (LGCP–SAC) | ✓ | ✓ | ✓ | ✓ | ✓ |

As shown in Table 1, prior works fail to address all requirements simultaneously. Our HBRL framework bridges this gap through: (i) LGCP-based Bayesian beliefs over unknown demand, (ii) PathMI non-myopic planning, (iii) dual-channel knowledge transfer for sample efficiency, and (iv) variance-normalized penalties for coordinated coverage.

## 3 System Model

### 3.1 System Description

As shown in Figure 1, we consider a spatial exploration task where $N$ autonomous agents are deployed to serve demand distributed across a two-dimensional service area $\mathcal{A} \subset \mathbb{R}^2$ of dimensions $L_x \times L_y$ meters. To enable tractable spatial modeling, the area is discretized into a uniform grid with spatial resolution $\Delta$ meters, yielding $G_x \times G_y$ cells, where $G_x = \lfloor L_x/\Delta \rfloor$ and $G_y = \lfloor L_y/\Delta \rfloor$.

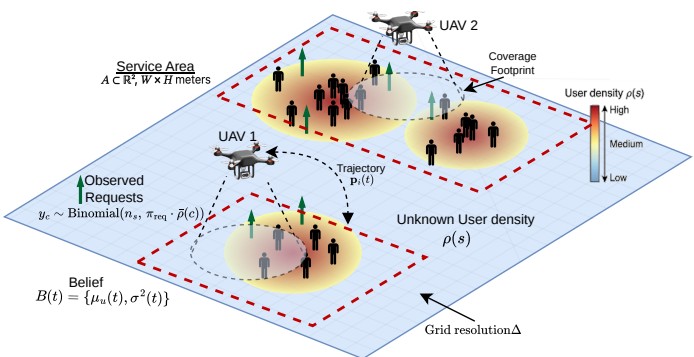

**Figure 1:** Multiple mobile agents spatial exploration over a discretized operational area with unknown demand, modeled via a spatial belief process. The illustrated scenario depicts UAVs as the instantiated agents, with further details provided in Section 4.

**Table 2:** Summary of Key Notations

| Symbol | Description | Symbol | Description | Symbol | Description |
|--------|-------------|--------|-------------|--------|-------------|
| $N$ | Number of agents | $T$ | Mission horizon | $\mathcal{A}$ | Service area |
| $\mathcal{G}$ | Discretized grid | $\Delta$ | Grid resolution | $G_x, G_y$ | Grid dimensions |
| $\mathbf{p}_i(t)$ | Position of agent $i$ | $d_{\max}$ | Max. displacement | $r_c$ | Sensing radius |
| $\mathcal{C}(t)$ | Covered cells at $t$ | $y_c(t)$ | Observed event count | $N_{\mathrm{req}}(t)$ | Total requests |
| $\rho(\mathbf{s})$ | Ground-truth density | $\bar{\rho}(c)$ | Cell-averaged density | $\boldsymbol{\mu}_j$ | Hotspot $j$ center |
| $\sigma_j$ | Hotspot $j$ spread | $\sigma_h$ | Hotspot diffusion rate | $r_b$ | Displacement bound |
| $\mathbf{u}$ | Log-intensity field | $\boldsymbol{\lambda}$ | Intensity field | $\sigma_c^2(t)$ | Posterior variance |
| $\tilde{\sigma}_c^2(t)$ | Predicted variance | $e_c(t)$ | Exposure indicator | $n_c$ | Visit count |
| $s_c$ | Staleness counter | $s_{\max}$ | Max. staleness | $\tau$ | Prior precision |
| $\beta$ | Spatial smoothness | $\gamma_v$ | Variance growth rate | $\sigma_{\max}^2$ | Max. variance |
| $r_g$ | Coverage radius in cells | $\sigma_{\min}^2$ | Min. variance | $L$ | Planning horizon |
| $D_{dir}$ | Candidate directions | $\xi_c$ | Diminishing return | $\epsilon$ | Stability constant |
| $m_c(t)$ | Overlap count | $K_w$ | Warm-start episodes | $\omega_1, \omega_2, \omega_3$ | Reward weights |
| $\omega_s$ | Staleness weight | $\gamma$ | Discount factor | $\tau_{\mathrm{soft}}$ | Soft update coeff. |

Each agent $i \in \{1, \ldots, N\}$ operates within a predefined operational subregion $\mathcal{A}_i \subset \mathcal{A}$. Subregions may overlap. Each agent senses demand within a circular footprint of radius $r_c$ meters, centered at position $\mathbf{p}_i(t) = (x_i(t), y_i(t))^\top$ at discrete time step $t$. All quantities indexed by $c \in \mathcal{G}$ are computed over the operational grid $\mathcal{G} = \bigcup_{i=1}^N \mathcal{G}_i$, the union of discretized subregions, rather than over the full service area.

User demand is assumed to be spatially heterogeneous and concentrated around $J$ hotspot regions whose locations and intensities are initially unknown to the agents. This assumption reflects realistic scenarios where event clusters around spatial hotspots whose structure must be discovered online. The mission objective is to jointly optimize agent trajectories over a horizon of $T$ time steps to maximize cumulative task reward while efficiently learning the underlying spatial demand distribution.

This setting requires balancing exploration of unknown regions against exploitation of discovered high-demand areas, a fundamental challenge in sequential decision-making under uncertainty (Krause & Guestrin, 2007). A summary of the key notations used throughout the paper is provided in Table 2.

### 3.2 Agent Dynamics Model

Each agent is modeled as a point mass entity with discrete-time dynamics. At each time step $t$, agent $i$ selects a displacement vector $\mathbf{v}_i(t) \in \mathbb{R}^2$. The position update is given by

$$\mathbf{p}_i(t+1) = \text{clip}(\mathbf{p}_i(t) + \mathbf{v}_i(t), \mathcal{A}), \tag{1}$$

where $\text{clip}(\cdot, \mathcal{A})$ enforces boundary constraints such that $0 \leq x_i(t) \leq L_x$ and $0 \leq y_i(t) \leq L_y$.

The control input for agent $i$ at time $t$ is specified as a continuous action $\mathbf{a}_i(t) \in [-1, 1]^2$, mapped to displacement via

$$\mathbf{v}_i(t) = \mathbf{a}_i(t) \cdot d_{\max}, \tag{2}$$

where $d_{\max}$ denotes the maximum displacement per time step. Since $\mathbf{a}_i(t) \in [-1, 1]^2$, this mapping enforces an $\ell_\infty$ bound on displacement, i.e. $\|\mathbf{v}_i(t)\|_\infty \leq d_{\max}$. For a fleet of $N$ agents, the joint action is $\mathbf{a}(t) = [\mathbf{a}_1(t)^\top, \ldots, \mathbf{a}_N(t)^\top]^\top \in [-1, 1]^{2N}$.

The sensing footprint of agent $i$ at time $t$ is defined as the set of grid cells within sensing range:

$$\mathcal{C}_i(t) = \{c \in \mathcal{G} : \|\mathbf{p}_c - \mathbf{p}_i(t)\|_2 \leq r_c\}, \tag{3}$$

where $\mathbf{p}_c \in \mathbb{R}^2$ denotes the center position of grid cell $c$. The joint coverage at time $t$ is $\mathcal{C}(t) = \bigcup_{i=1}^N \mathcal{C}_i(t)$.

### 3.3 Spatial Demand Model

Demand is modeled as a spatially heterogeneous density field over the operational area. The ground-truth normalized density $\rho : \mathcal{A} \to [0, 1]$ is generated as a mixture of $J$ Gaussian hotspots:

$$\rho(\mathbf{s}) = \frac{1}{\rho_{\max}} \sum_{j=1}^J \exp\left(-\frac{\|\mathbf{s} - \boldsymbol{\mu}_j\|_2^2}{2\sigma_j^2}\right), \tag{4}$$

where $\boldsymbol{\mu}_j \in \mathcal{A}$ and $\sigma_j > 0$ denote the center and spatial spread of hotspot $j$, and $\rho_{\max}$ is a normalization constant ensuring $\rho(\mathbf{s}) \in [0, 1]$. To model realistic local demand fluctuations, hotspot centers undergo a bounded random walk during each episode:

$$\boldsymbol{\mu}_j(t+1) = \text{clip}\left(\boldsymbol{\mu}_j(t) + \sigma_h \boldsymbol{\epsilon}_t, \, \boldsymbol{\mu}_j^{(0)} \pm r_b\right), \tag{5}$$

where $\sigma_h > 0$ is the diffusion rate, $\boldsymbol{\epsilon}_t \sim \mathcal{N}(\mathbf{0}, \mathbf{I})$ denotes isotropic Gaussian noise, $\boldsymbol{\mu}_j^{(0)}$ is the base hotspot position, and $r_b$ bounds the maximum displacement. At episode reset, hotspot centers return to their base positions with small perturbation, preserving spatial structure across episodes while introducing within-episode non-stationarity.

The mobile agents have no prior knowledge of the hotspot parameters $\{\boldsymbol{\mu}_j, \sigma_j\}_{j=1}^J$ or its position, necessitating online learning of the spatial demand pattern from partial observations.

*Remark 1 (Density vs. Belief):* The density $\rho(\mathbf{s})$ represents the ground-truth spatial distribution used to generate observations. The LGCP intensity $\lambda(\mathbf{s}) = \exp(u(\mathbf{s}))$ is the agents' learned belief over event rates. While these quantities differ in scale and interpretation, regions with higher $\rho(\mathbf{s})$ induce higher learned intensity in the belief model.

### 3.4 Observation Model

Agents can observe events only within their current sensing regions. At each time step $t$, for each grid cell $c \in \mathcal{C}(t)$, the observed event count is sampled as

$$y_c(t) \sim \text{Binomial}(n_s, \pi_{\text{req}} \bar{\rho}(c)), \tag{6}$$

where $n_s$ denotes the number of sampling trials, $\pi_{\mathrm{req}} \in (0,1]$ is the event, or request, probability, and $\bar{\rho}(c)$ is the average normalized density over cell $c$, computed as

$$\bar{\rho}(c) = \frac{1}{|\mathcal{A}_c|} \int_{\mathcal{A}_c} \rho(\mathbf{s}) \, d\mathbf{s}, \tag{7}$$

where $\mathcal{A}_c \subset \mathcal{A}$ denotes the spatial region corresponding to cell $c$ and $|\mathcal{A}_c| = \Delta^2$ is the cell area. Cells not covered at time $t$ yield no observations.

The total observed count at time $t$ is

$$N_{\mathrm{req}}(t) = \sum_{c \in \mathcal{C}(t)} y_c(t). \tag{8}$$

*Remark 2 (Likelihood Approximation):* While observations are generated from a binomial model, a Poisson likelihood is employed during belief inference. With small success probability ($\pi_{\mathrm{req}} \leq 0.05$), this approximation is well justified under classical Poisson limit results (Le Cam, 1960).

### 3.5 Log-Gaussian Cox Process Belief Model

We adopt a Log-Gaussian Cox Process (LGCP) for spatial demand estimation, following the formulation of Møller et al. (Møller et al., 1998) with grid-based Laplace-approximated inference (Rue et al., 2009). Our contribution lies in its online application under partial observability with temporal belief decay, and its integration with information-driven planning and reinforcement learning for coordinating multiple mobile agents.

#### 3.5.1 LGCP Formulation

Following standard LGCP notation (Diggle et al., 2013), the log-intensity is modeled as a Gaussian process:

$$\lambda(\mathbf{s}) = \exp(u(\mathbf{s})), \quad u(\mathbf{s}) \sim \mathcal{GP}(0, \mathcal{K}), \tag{9}$$

where $u : \mathcal{A} \to \mathbb{R}$ is the latent log-intensity field and $\mathcal{K}$ denotes a covariance kernel encoding spatial correlation. The exponential link ensures positive intensity values, while the Gaussian prior induces a spatially correlated uncertainty structure (Møller et al., 1998).

#### 3.5.2 GMRF Prior

For computational tractability on the discretized grid $\mathcal{G}$, a Gaussian Markov Random Field (GMRF) prior is imposed (Rue & Held, 2005), providing sparse precision matrices for efficient inference. Let $\mathbf{u} \in \mathbb{R}^{G_x \cdot G_y}$ denote the vectorized log-intensity field over all grid cells.

Following standard GMRF constructions (Rue & Held, 2005), the prior is specified through its precision matrix:

$$\mathbf{Q} = \tau \mathbf{I} + \beta \mathbf{L}_G, \tag{10}$$

where $\tau > 0$ controls prior precision (regularization strength), $\beta > 0$ governs spatial smoothness, $\mathbf{I}$ is the identity matrix, and $\mathbf{L}_G$ denotes the graph Laplacian encoding four-connected neighbor relationships on the grid. For a grid cell with four neighbors, the graph Laplacian has diagonal entries $[\mathbf{L}_G]_{cc} = 4$ (the node degree) and off-diagonal entries $[\mathbf{L}_G]_{cj} = -1$ for neighboring cells $j$. This formulation induces the prior distribution $p(\mathbf{u}) \propto \exp\left(-\frac{1}{2}\mathbf{u}^\top \mathbf{Q}\mathbf{u}\right)$, which is proper and positive definite for $\tau > 0$.

#### 3.5.3 Posterior Inference via Laplace Approximation

Given observations $\mathbf{y}(t) = \{y_c(t)\}_{c \in \mathcal{C}(t)}$ from currently covered cells, the posterior distribution over $\mathbf{u}$ combines the GMRF prior with the Poisson likelihood. The resulting posterior is analytically intractable due to the non-conjugate Poisson likelihood. We employ a Laplace approximation via Newton iterations (Dovers et al., 2023), yielding a Gaussian approximation centered at the posterior mode:

$$p(\mathbf{u}|\mathbf{y}(t)) \approx \mathcal{N}(\boldsymbol{\mu}_u, \boldsymbol{\Sigma}_u), \tag{11}$$

where $\boldsymbol{\mu}_u$ denotes the posterior mode and $\boldsymbol{\Sigma}_u = \mathbf{H}^{-1}$ is the posterior covariance, with $\mathbf{H}$ being the Hessian of the negative log-posterior.

To handle partial observability, we introduce a binary exposure variable $e_c(t) \in \{0, 1\}$ indicating whether cell $c$ is observed at time $t$:

$$e_c(t) = \mathbb{I}(c \in \mathcal{C}(t)), \tag{12}$$

where $\mathbb{I}(\cdot)$ is the indicator function. For cells not currently covered, $y_c(t) = 0$ by convention, with the exposure mask nullifying their likelihood contribution. The log-posterior at time $t$ then becomes:

$$\log p(\mathbf{u}|\mathbf{y}(t)) \propto \sum_{c \in \mathcal{G}} e_c(t) \left[ y_c(t) u_c - \exp(u_c) \right] - \frac{1}{2} \mathbf{u}^\top \mathbf{Q} \mathbf{u}, \tag{13}$$

where $e_c(t) = 1$ for currently covered cells and $e_c(t) = 0$ otherwise. The gradient and Hessian follow as:

$$\mathbf{g}(t) = \mathbf{e}(t) \odot (\mathbf{y}(t) - \boldsymbol{\lambda}) - \mathbf{Q}\mathbf{u}, \tag{14}$$
$$\mathbf{H}(t) = \mathrm{diag}(\mathbf{e}(t) \odot \boldsymbol{\lambda}) + \mathbf{Q}, \tag{15}$$

where $\boldsymbol{\lambda} = \exp(\mathbf{u})$ is the intensity vector, $\mathbf{e}(t)$ denotes the exposure vector at time $t$, and $\odot$ represents element-wise multiplication. The Newton update is:

$$\mathbf{u}^{(k+1)} = \mathbf{u}^{(k)} + \mathbf{H}(t)^{-1} \mathbf{g}(t). \tag{16}$$

The Newton iterations are warm-started from the previous Maximum A Posteriori (MA) estimate, $\mathbf{u}^{(0)} = \boldsymbol{\mu}_u(t-1)$, yielding an online MAP tracking procedure that encourages temporal persistence of the intensity field.

A preconditioned conjugate gradient (PCG) solver (Rue & Held, 2005) computes the Newton step $\mathbf{H}(t)^{-1}\mathbf{g}(t)$ without explicit matrix inversion. The Hessian admits diagonal entries:

$$[\mathbf{H}(t)]_{cc} = e_c(t)\lambda_c + Q_{cc}, \tag{17}$$

where $Q_{cc} = \tau + 4\beta$ for interior cells under the first-order GMRF structure. We use these diagonal entries as a preconditioner $\mathbf{M} = \mathrm{diag}(\mathbf{H})$, which exploits the sparsity induced by the GMRF prior and accelerates PCG convergence.

Computing exact marginal posterior variances $\mathrm{diag}(\mathbf{H}^{-1})$ requires sparse matrix inversion, which is prohibitive for large grids (Rue et al., 2009). Instead, we employ a diagonal Laplace approximation (Ritter et al., 2018; Kirkpatrick et al., 2017):

$$\sigma_c^2(t) \triangleq \frac{1}{[\mathbf{H}(t)]_{cc}}. \tag{18}$$

This surrogate ~~does not yield~~ sacrifices calibrated marginal variances but preserves relative uncertainty ordering across cells, which makes it sufficient for exploration incentives where ranking drives decision-making. For brevity, we refer to this quantity as the *posterior variance* in the remainder of the paper, noting that it corresponds to a diagonal Laplace approximation rather than the exact marginal variance. A direct comparison against the full marginal variance is tractable at small grid scales, but the cost of exact inversion renders such comparison impractical at the operational grid size in real world scenarios. The proxy is used because it preserves the relative uncertainty ordering on which exploration decisions depend.

For observed cells ($e_c(t) = 1$), higher intensity estimates yield smaller variance proxies, reflecting increased confidence. For unobserved cells ($e_c(t) = 0$), the likelihood contributes no curvature, and uncertainty evolution is governed solely by the temporal belief dynamics described next.

### 3.6 Temporal Belief Dynamics

In partially observable environments, the relevance of past observations degrades over time due to limited and intermittent sensing. We introduce a two-step belief update that separates uncertainty growth from observation-based refinement.

### 3.6.1 Step 1: Prediction (Uncertainty Growth)

At each time step, before incorporating new observations, we apply variance growth to all cells to reflect potential changes in the underlying demand pattern:

$$\tilde{\sigma}_c^2(t) = \min\left(\sigma_c^2(t^-) \cdot (1 + \gamma_v),\ \sigma_{\max}^2\right), \quad \forall c \in \mathcal{G}, \tag{19}$$

where $\sigma_c^2(t^-)$ denotes the posterior variance at the end of the previous time step $t - 1$, $\gamma_v > 0$ is the variance growth rate per time step, and $\sigma_{\max}^2$ represents the maximum variance corresponding to complete uncertainty.

### 3.6.2 Step 2: Update (Observation Incorporation)

After the prediction step, we incorporate observations from currently covered cells. For observed cells, the variance is overwritten by the posterior variance from LGCP inference; for unobserved cells, the predicted variance is retained:

$$\sigma_c^2(t^+) = \begin{cases} \max\left(\dfrac{1}{\lambda_c^* + \tau + 4\beta},\ \sigma_{\min}^2\right) & \text{if } c \in \mathcal{C}(t) \\ \tilde{\sigma}_c^2(t) & \text{otherwise} \end{cases}, \tag{20}$$

where $\lambda_c^* = \exp(u_c^*)$ is the converged posterior mode intensity and $\sigma_{\min}^2 > 0$ is a small floor preventing numerical instability in high-intensity regions.

This predict-then-update structure serves two purposes: (i) it acknowledges that demand patterns may evolve, making older observations less reliable, and (ii) it incentivizes revisitation of stale regions where belief confidence has degraded. The belief state (Kaelbling et al., 1998) at time $t$ is thus characterized by $\mathcal{B}(t) = \{\boldsymbol{\mu}_u(t), \boldsymbol{\sigma}^2(t^+)\}$.

*Remark 3 (Revisitation Incentive):* Under this formulation, revisiting a cell $c$ at time $t$ yields fresh observations $y_c(t)$ and resets the posterior variance to the observation-based value. This creates a natural incentive for revisitation: cells not recently observed accumulate uncertainty via Equation 19, making them attractive targets for exploration reward. Cells recently observed maintain low variance until staleness accumulates.

## 3.7 Reward Structure

The reward at time step $t$ balances three objectives: service quality (primary task), exploration, and coordination efficiency:

$$r_t = \omega_1 R_{\text{service}}(t) + \omega_2 R_{\text{explore}}(t) - \omega_3 C_{\text{coord}}(t), \tag{21}$$

where $\omega_1, \omega_2, \omega_3 > 0$ are objective weights.

### 3.7.1 Service Reward

The service reward captures the primary mission objective:

$$R_{\text{service}}(t) = N_{\text{req}}(t), \tag{22}$$

where $N_{\text{req}}(t)$ denotes the total observed event count at time $t$ as defined in Equation 8. This term incentivizes agents to position themselves over high-demand regions.

### 3.7.2 Exploration Reward

The exploration reward encourages belief improvement through uncertainty reduction:

$$R_{\text{explore}}(t) = \sum_{c \in \mathcal{C}(t)} \left(\tilde{\sigma}_c^2(t) - \sigma_c^2(t^+)\right), \tag{23}$$

where $\tilde{\sigma}_c^2(t)$ denotes the predicted variance (before observation) and $\sigma_c^2(t^+)$ denotes the updated posterior variance (after observation) at cell $c$.

**Table 3:** Comparison of overlap penalty strategies for a cell covered by two agents.

| Cell History | $\tilde{\sigma}_c^2(t)$ | Var.-Norm. | Fixed |
|---|---|---|---|
| Never visited | 1.0 | 0.0 | 1.0 |
| Recently visited | 0.1 | 0.9 | 1.0 |
| Stale (grown variance) | 0.5 | 0.5 | 1.0 |

This formulation creates an implicit incentive mechanism: cells with high predicted variance (either unvisited or stale) yield large uncertainty reduction upon observation; recently visited cells with low variance provide minimal gain. Consequently, agents are naturally drawn toward unexplored or stale regions without requiring added revisitation penalties.

### 3.7.3 Coordination Cost

The coordination cost penalizes inefficient trajectories through redundant coverage and excessive travel:

$$C_{\text{coord}}(t) = P_{\text{overlap}}(t) + d_{\text{travel}}(t), \tag{24}$$

where

$$d_{\text{travel}}(t) = \delta_t \sum_{i=1}^{N} \|\mathbf{p}_i(t) - \mathbf{p}_i(t-1)\|_2, \tag{25}$$

with $\delta_t > 0$ serving as a scaling constant for dimensional consistency.

To avoid overlapping coverage, we use a variance-normalized overlap penalty. For each grid cell $c \in \mathcal{G}$, let $m_c(t) = |\{i : c \in \mathcal{C}_i(t)\}|$ denote the number of agent coverage footprints that include cell $c$ at time $t$. The overlap penalty is then defined as

$$P_{\text{overlap}}(t) = \sum_{c:m_c(t)\geq 2} \left(1 - \frac{\tilde{\sigma}_c^2(t)}{\sigma_{\max}^2}\right), \tag{26}$$

where $\tilde{\sigma}_c^2(t)$ is the predicted variance (before incorporating current observations), and $\sigma_{\max}^2$ represents complete uncertainty.

*Remark 4 (Penalty Interpretation):* The use of predicted variance $\tilde{\sigma}_c^2(t)$ allows the penalty to adapt to belief state. In unexplored or stale regions ($\tilde{\sigma}_c^2(t) \approx \sigma_{\max}^2$), the penalty is near zero, permitting collaborative sensing. In recently observed regions ($\tilde{\sigma}_c^2(t) \approx \sigma_{\min}^2$), the penalty is high, discouraging redundant coverage. This adaptive behavior allows joint exploration when uncertainty is high, but persistent overlap is discouraged once variance is reduced.

A fixed overlap penalty applies uniform cost regardless of belief state and discourages all overlap equally, including beneficial joint exploration in high-uncertainty regions. Table 3 summarizes this difference.

The overlap penalty is triggered only when $m_c(t) \geq 2$. Single-agent revisitation penalty is governed by the exploration reward and service objective, allowing UAVs to revisit high-demand regions when beneficial.

### 3.7.4 Weight Selection

The default objective weights are set as $\omega_1 = 5.0,\quad \omega_2 = 0.5,$ and $\quad \omega_3 = 1.0$. The service weight reflects the primary mission objective, while exploration and coordination terms provide sufficient incentive for information gathering and efficient spatial distribution. Weight sensitivity analysis is included in Section 5.

### 3.8 Problem Formulation

We aim to maximize cumulative reward over a mission horizon of $T$ time steps by jointly optimizing agent trajectories $\mathcal{T} = \{\mathbf{p}_i(t)\}_{i,t}$:

$$\max_{\mathcal{T}} \quad \sum_{t=1}^{T} r_t, \tag{27a}$$

$$\text{s.t.} \quad \|\mathbf{v}_i(t)\|_{\infty} \leq d_{\max}, \quad \forall i, \forall t, \tag{27b}$$

$$\mathbf{p}_i(t) \in \mathcal{A}, \quad \forall i, \forall t, \tag{27c}$$

$$\mathbf{p}_i(0) = \mathbf{p}_i^{\text{init}}, \quad \forall i, \tag{27d}$$

where Equation 27b bounds the maximum displacement per step, Equation 27c ensures agents remain within the service area, and Equation 27d specifies initial positions.

The optimization problem stated in Equation 27 is an instance of informative path planning (IPP) under partial observability. The underlying objective of maximizing information gain through sequential sensing is NP-hard in general (Krause et al., 2008), and greedy approximations rely on submodularity that does not extend to the full reward structure. The addition of dynamic demand, continuous agent dynamics, and coordination between multiple mobile agents further precludes exact methods, motivating the use of reinforcement learning. However, as pure deep RL suffers from poor sample efficiency, we address this gap through HBRL framework that combines Bayesian spatial modeling with deep reinforcement learning via dual-channel knowledge transfer, detailed in Section 4.

## 4 Proposed Solution

### 4.1 Two-Phase HBRL Framework

We instantiate the general framework of Section 3 for multi-UAV wireless service provisioning, where agents correspond to UAVs deployed as aerial base stations at fixed altitude, serving spatially distributed users. The spatial demand field corresponds to wireless service requests, the sensing footprint maps to each UAV's circular communication coverage of radius $r_c$, and the event probability $\pi_{\text{req}}$ corresponds to the per-user service request probability.

Figure 2 summarizes the two-phases of HBRL framework. In Phase 1 (Exploration), UAVs execute information-driven trajectories guided by Pathwise Mutual Information (PathMI) planning while building spatial belief maps through LGCP inference. In Phase 2 (Exploitation), a Soft Actor-Critic (SAC) agent takes over trajectory control, initialized with knowledge transferred from Phase 1 through two complementary channels: (i) *belief state transfer* via the trained LGCP belief maps informing the RL state representation, and (ii) ~~behavioral knowledge transfer~~ *behavioral transfer* via expert demonstrations from the LGCP phase seeding the RL replay buffer.

This hybrid design addresses a fundamental limitation of pure reinforcement learning approaches: the need to learn both *where* high-value regions exist and *how* to navigate efficiently. We separate these two questions by using LGCP for spatial learning and RL for policy optimization. The framework achieves superior sample efficiency compared to learning both from scratch.

### 4.2 PathMI Information Score Planning

During Phase 1, UAV trajectories are guided by an uncertainty-driven planner that evaluates candidate paths based on expected uncertainty reduction. In contrast to myopic (one-step) planners that focus on maximizing immediate information gain (Krause & Guestrin, 2007), our approach considers multi-step lookahead to identify globally beneficial trajectories, following the receding-horizon paradigm common in model predictive control (Mayne et al., 2000). We use an uncertainty-weighted coverage score (incorporating variance and staleness) as a computationally efficient surrogate for mutual information, which we term PathMI.[2]

---

[2]A preliminary comparison of exploration strategies operating on the LGCP belief model is provided in Appendix A.

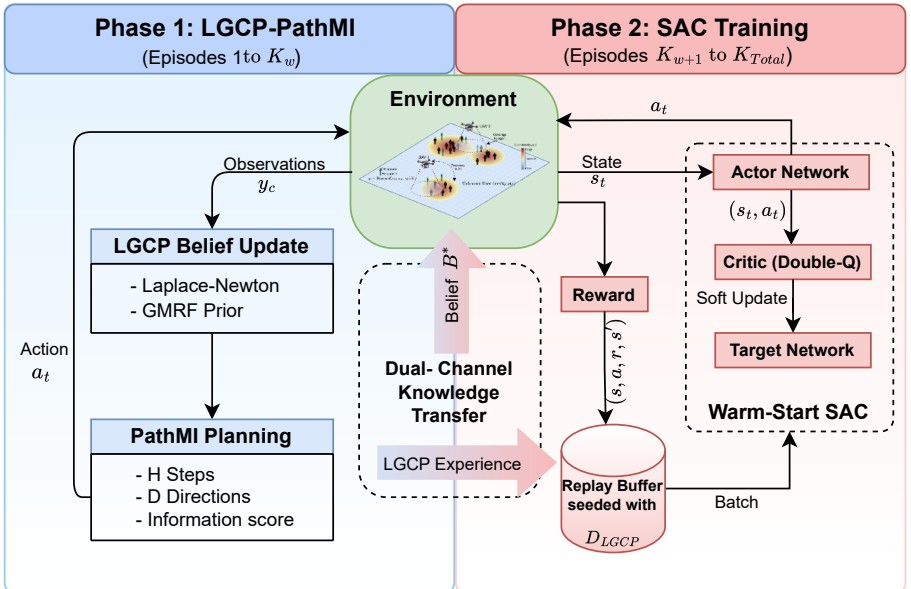

**Figure 2:** Overview of the proposed HBRL framework. Phase 1 employs LGCP-based belief inference and PathMI planning to guide information-driven exploration. Phase 2 performs policy optimization using Soft Actor-Critic (SAC), warm-started via dual-channel knowledge transfer: (i) ~~belief state initialization~~ belief state transfer and (ii) replay buffer seeding with LGCP-generated trajectories.

### 4.2.1 Candidate Path Generation

From each UAV's current grid position $(g_x, g_y)$, we generate $D_{dir} = 8$ candidate paths corresponding to the eight cardinal and diagonal directions. Each path extends for $L$ steps (the planning horizon), with positions computed as

$$\mathbf{p}^{(\ell)} = \text{clip}(\mathbf{p}_i + \ell \cdot d_{\max} \cdot \mathbf{d}, \mathcal{A}), \tag{28}$$

where $\mathbf{d} \in \{(\pm 1, 0), (0, \pm 1), (\pm 1, \pm 1)\}$ is the unit direction vector and $\ell \in \{1, \ldots, L\}$ indexes positions along the path. The resulting waypoints guide action selection, with actual displacement bounded by $d_{\max}$. The choice of straight-line paths balances computational efficiency against trajectory diversity; more complex path primitives could be substituted at increased computational cost.

### 4.2.2 PathMI Score Computation

For each candidate path $\mathcal{P}$, we compute the set of grid cells that would be observed along the trajectory:

$$\mathcal{O}(\mathcal{P}) = \bigcup_{\ell=1}^{L} \{c \in \mathcal{G} : \|\mathbf{p}_c - \mathbf{p}^{(\ell)}\|_2 \leq r_c\}, \tag{29}$$

where $\mathbf{p}^{(\ell)}$ denotes the world-coordinate position of waypoint $\ell$ along path $\mathcal{P}$. The PathMI Information Score aggregates uncertainty over cells along the path:

$$\text{PathMI}(\mathcal{P}) = \sum_{c \in \mathcal{O}(\mathcal{P})} \left[ \xi_c \cdot \sigma_c^2 + \omega_s \cdot \min\left(\frac{s_c}{s_{\max}}, 1\right) \cdot \sigma_c^2 \right], \tag{30}$$

where $\sigma_c^2$ is the current posterior variance, $s_c$ is the staleness (time steps since last observation) at cell $c$, $s_{\max}$ is a normalization constant equal to $T$, and $\omega_s$ is the staleness weighting coefficient. The staleness term encourages revisitation of regions where belief confidence has degraded. The factor $\xi_c$ down-weights

cells with repeated observations, reflecting the diminishing returns characteristic of information gain (Krause et al., 2008):

$$\xi_c = \frac{1}{1 + n_c}, \tag{31}$$

where $n_c \geq 0$ denotes the cumulative observation count at cell $c$. Unvisited cells ($n_c = 0$) yield $\xi_c = 1$, providing maximum incentive for first-time exploration, while previously visited cells yield progressively smaller factors as observations accumulate. This formulation ensures that PathMI prioritizes novel regions while still permitting revisitation when staleness is high.

*Remark 5:* The PathMI score (Equation 30) approximates mutual information (Krause & Guestrin, 2007) through variance aggregation. Under Gaussian assumptions, high-variance regions yield greater entropy reduction upon observation, making cumulative variance a computationally efficient surrogate for ranking candidate paths by expected information gain (MacKay, 1992; Riis et al., 2022). True mutual information computation requires marginalization over future observations, which is computationally prohibitive for online planning.

### 4.2.3 PathMI Action selection

The UAV agent selects the path with maximum PathMI score:

$$\mathcal{P}^* = \arg \max_{\mathcal{P} \in \{\mathcal{P}_1, \ldots, \mathcal{P}_D\}} \text{PathMI}(\mathcal{P}). \tag{32}$$

The action is then computed as the normalized direction toward the first waypoint of $\mathcal{P}^*$:

$$\mathbf{a}_i = \frac{\Delta \mathbf{p}}{\max(\|\Delta \mathbf{p}\|_2, \epsilon)}, \tag{33}$$

where $\Delta \mathbf{p} = \mathbf{p}_{\mathcal{P}^*(1)} - \mathbf{p}_i$ denotes the displacement to the first waypoint, $\mathbf{p}_{\mathcal{P}^*(1)}$ is the world position of that waypoint, and $\epsilon > 0$ is a small constant (e.g., $\epsilon = 10^{-6}$) preventing division by zero when the waypoint coincides with the current position after clipping. The action is then clipped to $[-1, 1]^2$ before execution. This receding-horizon approach re-plans at each step, adapting to updated beliefs while maintaining non-myopic trajectory structure.

## 4.3 Dual-Channel Knowledge Transfer

In contrast to prior approaches that transfer either state representations (Zhu et al., 2023) or behavioral demonstrations (Hester et al., 2018), but not both simultaneously, our framework transfers both channels, enabling complementary benefits.

### 4.3.1 Channel 1: Belief State Transfer

At the conclusion of Phase 1, the LGCP belief state $\mathcal{B}^* = \{\boldsymbol{\mu}_u^*, \boldsymbol{\sigma}^{2*}\}$ encodes the spatial demand information learned during information-driven exploration. This belief is transferred to Phase 2 by initializing the reinforcement learning environment at the start of training with $\mathcal{B}_{\text{RL}}(0) \leftarrow \mathcal{B}^*$. Belief state transfer serves as an initialization mechanism that provides an informed prior for early policy gradient updates, mitigating instability associated with uninformed exploration during the initial stages of training. To preserve episodic independence and avoid cross-episode accumulation of spatial information, subsequent episodes reset the belief state to the uninformative prior $\mathcal{B}_0 = \{\mathbf{0}, \mathbf{1}\}$. Learning progress is retained in the policy parameters and replay buffer rather than in the belief state itself.

### 4.3.2 Channel 2: Behavioral Transfer

During Phase 1, the LGCP planner generates high-quality trajectories optimized for information acquisition. We store these trajectories as demonstration data:

$$\mathcal{D}_{\text{LGCP}} = \{(\mathbf{s}_t, \mathbf{a}_t, r_t, \mathbf{s}_{t+1}, d_t)\}_{t=1}^{T_w}, \tag{34}$$

where $T_w = K_w \times T$ is the total number of transitions collected over $K_w$ warm-start episodes, and $d_t \in \{0, 1\}$ is the terminal flag indicating whether $\mathbf{s}_{t+1}$ is a terminal state. Following the learning from demonstrations paradigm (Hester et al., 2018), these transitions seed the SAC replay buffer before training begins:

$$\mathcal{D}_{\text{SAC}} \leftarrow \mathcal{D}_{\text{LGCP}}. \tag{35}$$

This seeding provides two benefits: (i) the agent can immediately sample informative transitions for gradient updates without random exploration, and (ii) the demonstrations encode implicit coordination patterns that accelerate multi-UAV coordination learning.

The two channels address different aspects of the learning problem. The belief channel provides uncertainty estimates for reward shaping (overlap penalty) and global belief summaries indicating overall exploration progress. The behavioral channel provides state-action pairs demonstrating effective spatial exploration patterns. Ablation studies in Section 5 quantify the individual and combined contributions of each channel, demonstrating that full dual-channel transfer outperforms either channel alone.

The complementary effect can be interpreted through early state-distribution alignment and reward calibration. The replay buffer $\mathcal{D}_{\text{LGCP}}$ contains transitions collected under the trained belief $\mathcal{B}^*$, where the state summary $\boldsymbol{\phi} = [\bar{\lambda}, \bar{\sigma}^2, \bar{n}_{\text{obs}}]^\top$ (cf. Equation 37) reflects structured spatial information. If SAC instead begins from the uninformed prior $\mathcal{B}_0$, the initial on-policy state distribution over $\boldsymbol{\phi}$ differs from that of the buffered demonstrations, creating a distribution mismatch during early critic updates. Initializing the first SAC episode with $\mathcal{B}^*$ can reduce this mismatch, improving alignment between replayed and current transitions at the start of training. Moreover, both the exploration reward and the variance-normalized overlap penalty depend on calibrated posterior uncertainty. Under $\mathcal{B}_0$, posterior variance is initially uniform, reducing reward contrast across actions. Initializing with $\mathcal{B}^*$ restores spatial structure at the start of Phase 2, strengthening reward contrast across action.

*Remark 6 (Distribution mismatch impacts off-policy estimates):* Let $d_\pi(s)$ be the discounted state visitation distribution under policy $\pi$ and $d_\mathcal{D}(s)$ the empirical state distribution induced by samples from replay buffer $\mathcal{D}$. For any bounded function $f : \mathcal{S} \to \mathbb{R}$,

$$|\mathbb{E}_{s \sim d_\pi}[f(s)] - \mathbb{E}_{s \sim d_\mathcal{D}}[f(s)]| \leq \|f\|_\infty \, \mathrm{V}(d_\pi, d_\mathcal{D}), \tag{36}$$

where $\mathrm{V}(\cdot, \cdot)$ denotes total variation distance. This follows directly from the definition of total variation distance, $\mathrm{V}(P, Q) = \sup_A |P(A) - Q(A)|$. Hence, when replay and on-policy state distributions differ, quantities learned under $d_\mathcal{D}$ (e.g., value estimates) may generalize poorly to states encountered under $d_\pi$.

In our setting, $\mathcal{D}_{\text{LGCP}}$ contains transitions collected under belief $\mathcal{B}^*$, inducing a structured distribution over the state summary $\boldsymbol{\phi}$. Initializing the first SAC episode with $\mathcal{B}^*$ reduces the initial mismatch relative to starting from $\mathcal{B}_0$, thereby improving the relevance of early critic updates.

### 4.3.3 Episode Reset Policy

At the start of each SAC episode in Phase 2, the LGCP belief state is reset to the uninformed prior $\mathcal{B}_0 = \{\mathbf{0}, \mathbf{1}\}$. This prevents accumulation of spatial information across episodes, ensuring that each episode begins from an unknown demand state. Learning progress is retained in the policy parameters and replay buffer, not in the belief state. The transferred belief $\mathcal{B}^*$ is used only to initialize the *first* SAC episode, providing a non-trivial starting point for initial policy gradient updates.

### 4.4 MDP Formulation

The multi-UAV coordination problem is a partially observable Markov decision process (POMDP), as agents observe demand only within their sensing footprint. Following the belief-space Markov decision process (MDP) formulation (Kaelbling et al., 1998), we implement this as an MDP over compressed observations. The unknown demand field is tracked internally through LGCP-based belief updates, and the policy receives low-dimensional belief summaries as state input. The environment maintains the full belief for state evolution and reward computation. Rationale and implications are discussed below in Remark 7.

### 4.4.1 State Space

The state vector $\mathbf{s}_t \in \mathbb{R}^{3N+4}$ comprises:

$$\mathbf{s}_t = \left[\tilde{\mathbf{p}}_1, \ldots, \tilde{\mathbf{p}}_N, \boldsymbol{\phi}, \bar{c}_1, \ldots, \bar{c}_N, \frac{t}{T}\right]^\top, \tag{37}$$

where $\tilde{\mathbf{p}}_i = 2(\mathbf{p}_i/[L_x, L_y]^\top) - \mathbf{1} \in [-1, 1]^2$ denotes the normalized UAV positions ($2N$ dimensions), $\boldsymbol{\phi} = [\bar{\lambda}, \bar{\sigma}^2, \bar{n}_{\text{obs}}]^\top \in \mathbb{R}^3$ represents the global belief summary and is computed as arithmetic mean intensity, mean variance, mean observation count over the operational regions assigned to the UAVs, rather than over the entire grid, $\bar{c}_i \in [0, 1]$ indicates coverage progress for UAV $i$ ($N$ dimensions), and $t/T \in [0, 1]$ is the normalized time step.

The coverage progress $\bar{c}_i$ for UAV $i$ is computed as the fraction of grid cells visited by UAV $i$ up to time $t$:

$$\bar{c}_i(t) = \frac{1}{|\mathcal{G}_i|} \sum_{c \in \mathcal{G}_i} \mathbb{I}\left(\exists \tau \le t : c \in \mathcal{C}_i(t')\right), \tag{38}$$

*Remark 7:* We intentionally use compact global belief summaries rather than the full belief grid to keep the state low-dimensional and scalable. While this compression sacrifices local spatial detail (the agent knows "overall uncertainty is high" but not "where"), it ensures stable training and $\mathcal{O}(1)$ state growth with grid size. This design reflects a deliberate trade-off between representational fidelity and scalability. The UAV positions provide explicit spatial grounding, and the smoothness induced by the GMRF prior mitigates sharply distinct spatial configurations during practical exploration. Empirical results in Section 5.2 confirm that the compressed state retains sufficient decision-relevant structure for stable and improved performance.

### 4.4.2 Action Space

The joint action $\mathbf{a}_t \in [-1, 1]^{2N}$ specifies bounded motion commands for all UAVs:

$$\mathbf{a}_t = [\mathbf{a}_1^\top, \ldots, \mathbf{a}_N^\top]^\top, \quad \mathbf{a}_i \in [-1, 1]^2. \tag{39}$$

Each component represents a normalized displacement direction, mapped to per-step motion via $\mathbf{v}_i = \mathbf{a}_i \cdot d_{\max}$ as defined in Equation 2.

### 4.4.3 Reward Function

The reward $r_t$ follows the three-objective structure defined in Equation 21, balancing service quality ($R_{\text{service}}$), exploration incentive ($R_{\text{explore}}$), and coordination efficiency ($C_{\text{coord}}$). In the wireless service provisioning context, these terms may map to established operational objectives: $R_{\text{service}}$ aligns with throughput optimisation (Wu et al., 2018) and user quality of service (Hao et al., 2024); $R_{\text{explore}}$ aligns with reducing demand uncertainty, a standard objective in information-driven planning (Krause et al., 2008); and $C_{\text{coord}}$ aligns with spectrum-contention avoidance (Rizvi & Boyle, 2025) and UAV energy budget (Zeng et al., 2019). Following Rizvi & Boyle (2025), we deliberately abstract from application-specific models (wireless channel modelling, UAV propulsion energy, flight dynamics) to focus on the framework-level contribution.

### 4.4.4 Transition Dynamics

State transitions follow the deterministic dynamics in Equation 1, with stochasticity arising from the binomial observation model (Equation 6) and the LGCP belief update.

### 4.5 Soft Actor-Critic Training

The continuous $2N$-dimensional action space precludes discrete-action methods such as DQN (Mnih et al., 2015). Among continuous-action algorithms, we select Soft Actor-Critic (SAC) (Haarnoja et al., 2018a) for three reasons: (i) its off-policy nature enables replay buffer seeding with LGCP demonstrations, whereas

on-policy methods like Proximal Policy Optimization (PPO) (Schulman et al., 2017) would discard these transitions; (ii) entropy regularization maintains exploration during the transition from demonstration-driven to self-generated experience, preventing premature convergence; and (iii) clipped double Q-learning provides greater stability than Deep Deterministic Policy Gradient (DDPG) (Lillicrap et al., 2016) in joint action settings for multiple agents.

### 4.5.1 Maximum Entropy Objective

SAC optimizes a maximum entropy objective (Haarnoja et al., 2018a):

$$J(\pi) = \mathbb{E}_{\tau \sim \pi} \left[ \sum_{t=0}^{T} \gamma^t \left( r(\mathbf{s}_t, \mathbf{a}_t) + \alpha \mathcal{H}(\pi(\cdot|\mathbf{s}_t)) \right) \right], \tag{40}$$

where $\gamma \in (0, 1)$ denotes the discount factor, $\alpha > 0$ is the temperature parameter controlling exploration-exploitation balance, and $\mathcal{H}(\cdot)$ represents entropy. The entropy regularization encourages exploration during early training while allowing convergence to near-deterministic policies as learning progresses.

### 4.5.2 SAC Architecture

Following standard practice (Haarnoja et al., 2018a), the policy and value networks use multi-layer perceptrons (MLPs) with two hidden layers of 256 units each and ReLU activations. The policy network $\pi_\theta$ outputs mean $\boldsymbol{\mu}_\theta(\mathbf{s})$ and log-standard-deviation $\log \boldsymbol{\sigma}_\theta(\mathbf{s})$ for a squashed Gaussian distribution:

$$\mathbf{a} = \tanh \left( \boldsymbol{\mu}_\theta(\mathbf{s}) + \boldsymbol{\sigma}_\theta(\mathbf{s}) \odot \boldsymbol{\epsilon} \right), \quad \boldsymbol{\epsilon} \sim \mathcal{N}(\mathbf{0}, \mathbf{I}), \tag{41}$$

where $\tanh(\cdot)$ bounds actions to $[-1, 1]^{2N}$. Two Q-networks $Q_{\phi_1}, Q_{\phi_2}$ with separate target networks $Q_{\bar{\phi}_1}, Q_{\bar{\phi}_2}$ mitigate overestimation bias through clipped double Q-learning (Fujimoto et al., 2018).

### 4.5.3 Loss Functions

The Q-networks are trained by minimizing the soft Bellman residual:

$$\mathcal{L}_Q(\phi_j) = \mathbb{E}_{(\mathbf{s}, \mathbf{a}, r, \mathbf{s}', d) \sim \mathcal{D}} \left[ \left( Q_{\phi_j}(\mathbf{s}, \mathbf{a}) - y \right)^2 \right], \tag{42}$$

where the target $y$ is computed as:

$$y = r + \gamma(1 - d) \left( \min_{j=1,2} Q_{\bar{\phi}_j}(\mathbf{s}', \tilde{\mathbf{a}}') - \alpha \log \pi_\theta(\tilde{\mathbf{a}}'|\mathbf{s}') \right), \tag{43}$$

with $\tilde{\mathbf{a}}' \sim \pi_\theta(\cdot|\mathbf{s}')$ and $d \in \{0, 1\}$ indicating terminal states. The policy is updated by maximizing the expected return with entropy bonus:

$$\mathcal{L}_\pi(\theta) = \mathbb{E}_{\mathbf{s} \sim \mathcal{D}, \tilde{\mathbf{a}} \sim \pi_\theta} \left[ \alpha \log \pi_\theta(\tilde{\mathbf{a}}|\mathbf{s}) - \min_{j=1,2} Q_{\phi_j}(\mathbf{s}, \tilde{\mathbf{a}}) \right]. \tag{44}$$

The temperature $\alpha$ is automatically tuned to maintain a target entropy $\bar{\mathcal{H}}$ (Haarnoja et al., 2018b):

$$\mathcal{L}_\alpha = \mathbb{E}_{\tilde{\mathbf{a}} \sim \pi_\theta} \left[ -\alpha \left( \log \pi_\theta(\tilde{\mathbf{a}}|\mathbf{s}) + \bar{\mathcal{H}} \right) \right]. \tag{45}$$

### 4.5.4 Target Network Updates

Target networks are updated via exponential moving average (Polyak averaging) (Lillicrap et al., 2016):

$$\bar{\phi}_j \leftarrow \tau_{\text{soft}} \phi_j + (1 - \tau_{\text{soft}}) \bar{\phi}_j, \tag{46}$$

where $\tau_{\text{soft}} \ll 1$ ensures stable target values.

The complete framework is presented in Figure 3. Algorithm 1 details the LGCP-PathMI exploration phase, while Algorithm 2 describes the warm-started SAC training phase.

*Remark 8:* Integration of LGCP with reinforcement learning for spatial exploration and service provisioning is less explored. Our *methodological* contribution lies in the dual-channel transfer mechanism and the variance-normalized coordination penalties, building upon the LGCP or SAC components.

**Algorithm 1: Phase 1 − LGCP–PathMI Exploration**

1: Initialize log-intensity $\mathbf{u} \leftarrow \mathbf{0}$, variance $\boldsymbol{\sigma}^2 \leftarrow \mathbf{1}$
2: Initialize observation counts $\mathbf{n} \leftarrow \mathbf{0}$, staleness $\mathbf{s} \leftarrow \mathbf{0}$
3: Initialize precision matrix $\mathbf{Q} = \tau\mathbf{I} + \beta\mathbf{L}_G$
4: Initialize demonstration buffer $\mathcal{D}_{\text{LGCP}} \leftarrow \emptyset$
5: **for** episode $k = 1$ to $K_w$ **do**
6:     Reset environment; reset belief to prior $\mathcal{B}_0$
7:     Reset staleness $\mathbf{s} \leftarrow \mathbf{0}$, observation counts $\mathbf{n} \leftarrow \mathbf{0}$
8:     **for** $t = 0$ to $T - 1$ **do**
9:         Observe state $\mathbf{s}_t$
10:         Update staleness: $s_c \leftarrow s_c + 1$ for all $c \in \mathcal{G}$
11:         **for** each UAV $i = 1$ to $N$ **do**
12:             Generate $D_{dir}$ candidate paths Equation 28
13:             Compute PathMI scores Equation 30
14:             Select $\mathcal{P}^* = \arg\max_d \text{PathMI}(\mathcal{P}_d)$
15:             Set $\mathbf{a}_i$ toward first waypoint Equation 33
16:         **end for**
17:         Execute joint action $\mathbf{a}_t$, observe $\mathbf{y}(t)$
18:         Set exposure: $e_c(t) \leftarrow \mathbb{I}(c \in \mathcal{C}(t))$ for all $c$
19:         Update observation counts: $n_c \leftarrow n_c + e_c(t)$ for all $c$
20:         Apply variance growth Equation 19; store $\tilde{\sigma}_c^2(t)$
21:         Update LGCP belief Equation 16
22:         Apply variance update Equation 20 $\rightarrow \sigma_c^2(t^+)$
23:         Compute reward $r_t$ Equation 21 using $\tilde{\sigma}_c^2(t)$ and $\sigma_c^2(t^+)$
24:         Compute next state $\mathbf{s}_{t+1}$
25:         Set terminal flag $d_t \leftarrow \mathbb{I}(t = T - 1)$
26:         Reset staleness: $s_c \leftarrow 0$ for all $c \in \mathcal{C}(t)$
27:         Store $(\mathbf{s}_t, \mathbf{a}_t, r_t, \mathbf{s}_{t+1}, d_t)$ in $\mathcal{D}_{\text{LGCP}}$
28:     **end for**
29: **end for**
30: Set final belief $\mathcal{B}^* = \{\boldsymbol{\mu}_u, \boldsymbol{\sigma}^2\}$
31: **return** $\mathcal{B}^*$, $\mathcal{D}_{\text{LGCP}}$

**Algorithm 2: Phase 2 − Warm-Started SAC Training**

1: Initialize replay buffer $\mathcal{D} \leftarrow \mathcal{D}_{\text{LGCP}}$     ▷ Buffer seeding
2: Initialize actor $\pi_\theta$, critics $Q_{\phi_1}, Q_{\phi_2}$
3: Initialize targets $\bar{Q}_{\phi_1} \leftarrow Q_{\phi_1}, \bar{Q}_{\phi_2} \leftarrow Q_{\phi_2}$
4: Initialize temperature $\alpha$
5: **for** episode $k = K_w + 1$ to $K$ **do**
6:     **if** $k = K_w + 1$ **then**
7:         Initialize belief with $\mathcal{B}^*$ from Phase 1
8:     **else**
9:         Reset belief to prior $\mathcal{B}_0 = \{\mathbf{0}, \mathbf{1}\}$
10:     **end if**
11:     Reset staleness $\mathbf{s} \leftarrow \mathbf{0}$, observation counts $\mathbf{n} \leftarrow \mathbf{0}$
12:     **for** $t = 0$ to $T - 1$ **do**
13:         Observe $\mathbf{s}_t$, sample $\mathbf{s}_t \sim \pi_\theta(\cdot|s_t)$
14:         Execute $\mathbf{a}_t$, observe $y(t)$
15:         Apply variance growth Equation 19 $\rightarrow \tilde{\sigma}_c^2(t)$
16:         Update LGCP belief via Equation 16
17:         Apply variance update Equation 20 $\rightarrow \sigma_c^2(t^+)$
18:         Compute reward $r_t$ via Equation 21
19:         Compute next state $\mathbf{s}_{t+1}$
20:         Set terminal flag $d_t \leftarrow \mathbb{I}(t = T - 1)$
21:         Store $(\mathbf{s}_t, \mathbf{a}_t, r_t, \mathbf{s}_{t+1}, d_t)$ in $\mathcal{D}$
22:         Sample mini-batch from $\mathcal{D}$
23:         Update critics via Equation 42, actor via Equation 44
24:         Update $\alpha$ via Equation 45
25:         Soft-update targets via Equation 46
26:     **end for**
27: **end for**
28: **return** trained policy $\pi_\theta$

**Figure 3:** Two-phase training pipeline: LGCP–PathMI exploration (left) and warm-started SAC training (right).

## 4.6 Complexity Analysis

### 4.6.1 Time Complexity

The LGCP belief update via Laplace-Newton requires $\mathcal{O}(n_{\text{Newton}} \cdot n_{\text{PCG}} \cdot |\mathcal{G}|)$ operations per step, where $n_{\text{Newton}}$ and $n_{\text{PCG}}$ denote the number of Newton and PCG iterations, respectively, and $|\mathcal{G}| = G_x \times G_y$ is the union of operational grid size. PathMI planning evaluates $D_{dir}$ candidate paths, each covering $\mathcal{O}(L \cdot r_g^2)$ cells where $r_g = \lceil r_c/\Delta \rceil$ is the coverage radius in grid cells, yielding $\mathcal{O}(D \cdot L \cdot r_g^2)$ per UAV per step. SAC updates are $\mathcal{O}(B \cdot d_{\text{net}})$ where $B$ is the batch size and $d_{\text{net}}$ denotes the network parameter count.

### 4.6.2 Space Complexity

The LGCP belief requires $\mathcal{O}(|\mathcal{G}|)$ storage for mean, variance, and exposure fields. The replay buffer stores $\mathcal{O}(|\mathcal{D}| \cdot d_s)$ transitions where $d_s = 3N + 4$ is the state dimension. SAC networks require $\mathcal{O}(d_{\text{net}})$ parameters.

### 4.6.3 Scalability

The framework scales linearly with the number of UAVs $N$ in state dimension $(3N+4)$ and action dimension $(2N)$. For moderate fleet sizes, the joint policy formulation provides sample-efficient implicit coordination through shared belief state. Learning by parameter-sharing or other techniques for very large fleet sizes remains an open challenge for future work. For very large grids, the GMRF structure ensures sparse precision matrices, maintaining computational tractability.

**Table 4:** Simulation and Algorithm Parameters

| Parameter | Value | Parameter | Value |
|---|---|---|---|
| Service area ($\mathcal{A}$) | $2000 \times 2000$ m$^2$ | Hidden layers | 256–256 |
| Grid resolution ($\Delta$) | 20 m | Learning rate ($\eta$) | $3 \times 10^{-4}$ |
| Grid size ($G_x \times G_y$) | $100 \times 100$ | Replay buffer size ($|\mathcal{D}|$) | $10^4$ |
| UAVs ($N$) | 2–4 | Batch size ($B$) | 256 |
| Coverage radius ($r_c$) | 100 m | Discount factor ($\gamma$) | 0.99 |
| Max displacement ($d_{\max}$) | 15 m | Soft update ($\tau_{\text{soft}}$) | 0.005 |
| No. of Hotspots ($J$) | 3–5 | Target entropy ($\bar{\mathcal{H}}$) | $-\dim(\mathcal{A})$ |
| Hotspot Diffusion rate ($\sigma_h$) | 1 m/step | Learning starts | 100 steps |
| Displacement bound ($r_b$) | 75 m | Total episodes ($K$) | 200 |
| Request prob. ($\pi_{\text{req}}$) | 0.05 | Warm-start episodes ($K_w$) | 10–50 |
| Smoothness ($\beta$) | 0.2 | Episode length ($T$) | 200 |
| Variance growth ($\gamma_v$) | 0.002 | Planning horizon ($L$) | 5 |
| Variance bounds ($\sigma^2_{\min/\max}$) | 0.01 / 1.0 | Candidate directions ($D_{dir}$) | 8 |
| Staleness weight ($\omega_s$) | 0.1 | Service weight ($\omega_1$) | 5.0 |
| Prior precision ($\tau$) | 1.0 | Exploration weight ($\omega_2$) | 0.5 |
| Sampling trials ($n_s$) | 100 | Coordination weight ($\omega_3$) | 1.0 |
| Newton iterations ($n_{\text{Newton}}$) | 3 | PCG iterations ($n_{\text{PCG}}$) | 8 |
| Max staleness ($s_{\max}$) | $T$ | | |

## 5 Experimental Evaluation

### 5.1 Experimental Setup

We perform simulations on an Apple M1 Pro with 3.2 GHz CPU, 16 GB RAM, and 16-core GPU, programmed in Python using Stable-Baselines3 for SAC implementation. The service area is $2000 \times 2000$ m$^2$ discretized into grids with resolution $\Delta = 20$ m, with $N = 2$ UAVs operating at fixed altitude with a coverage radius $r_c = 100$ m. $r_c = 5\Delta$ ensures that each UAV footprint spans multiple grid cells and avoids discretization artefacts observed when $r_c < \Delta$. User demand is generated as a mixture of $J = 3-5$ Gaussian hotspots with randomly sampled centers and spreads. All results are averaged over 5 independent seeds. We compare the proposed HBRL against the following baselines: Pure LGCP-PathMI (no RL), Pure SAC (cold start), SAC + Belief Transfer Only, and SAC + Buffer Seeding Only. All methods use identical SAC parameters summarized in Table 4. Values for SAC hyper-parameters follow the standard recommendations from the original SAC implementation (Haarnoja et al., 2018a). Unless stated otherwise, default parameters are used throughout.

As an imitation-based warm-start baseline, we include Behavior Cloning (BC) followed by reinforcement learning. After Phase 1 the SAC actor is first pre-trained offline via supervised learning to minimize negative log-likelihood over Phase 1 PathMI state-action pairs, using the same compressed state representation as the proposed method. After pre-training, the imitation objective is removed and the policy is fine-tuned using standard SAC updates with an empty replay buffer. This baseline uses identical Phase 1 data as our approach but incorporates it through actor pre-training rather than replay buffer seeding, allowing direct comparison of demonstration utilization strategies.

We evaluate our proposed framework through a series of experiments designed to assess learning efficiency, robustness, and scalability. Section 5.2 compares learning performance against baselines. Sections 5.3 and 5.4 examine the sensitivity to warm-start duration, planning horizon, and fleet size. Section 5.5 analyzes belief dynamics and uncertainty calibration, and Section 5.6 examines robustness to experience loss.

## 5.2 Learning Performance and Efficiency

Figure 4 compares learning behaviour across all methods. Pure LGCP maintains stable reward without policy adaptation, while Pure RL begins with the lowest reward due to random exploration before gradually improving. BC follows the LGCP reward levels during Phase 1 and transitions smoothly without the dip exhibited by HBRL, yet converges to similar asymptotic reward as Pure RL, indicating that action imitation accelerates early learning but does not improve final policy quality. HBRL eventually outperforms both Pure RL and BC+SAC, improving over Pure RL by 10.8%, and reaching its final reward level 38% earlier. Despite the compressed belief, learning through HBRL remains stable and outperforms both baselines, suggesting the summary retains sufficient structure. The gap between BC+SAC and HBRL demonstrates that replay buffer seeding provides benefits beyond behavioral imitation.

While reward curves capture overall task performance, it is equally important to examine how each method influences the quality of the learned belief state, since accurate estimation of user demand is fundamental to the requirements of service providers. Figure 5 shows the evolution of average posterior variance (diagonal Laplace proxy), where lower values indicate higher confidence in the inferred demand field. Pure LGCP maintains a nearly constant variance of approximately 0.66 because each episode begins with a fresh belief map, preventing long-term accumulation of information. Pure RL, in contrast, steadily reduces variance over time, converging to 0.37 by episode 155 as the agent revisits informative regions during policy improvement. HBRL, whereby starting learning later than Pure-RL, achieves the fastest and best variance reduction, matching best Pure-RL performance 32% faster. It also reduces the variance against the benchmark by 11%.

Figure 6 shows the ablation study for dual-channel knowledge transfer. Belief transfer alone provides negligible improvement (+0.6% reward, 3% fewer episodes to reach Pure RL reward), indicating that spatial knowledge informs *where* uncertainty exists but not *how* to act. Buffer seeding alone yields substantial gains (+7.7% reward, 33% fewer episodes to reach Pure RL reward) by providing LGCP-demonstrated trajectories that accelerate policy learning. Notably, combining both channels achieves highest reward improvement and 38% fewer episodes to reach Pure RL reward. This demonstrates that while belief transfer has minimal effect in isolation, it provides meaningful benefit when used in conjunction with replay buffer seeding. The spatial context helps the agent generalize beyond the specific trajectories in the buffer, enabling more effective exploration of high-value regions identified by the LGCP posterior.

## 5.3 Warm start and Planning Horizon Ablations

The warm-start ablation in Figure 7 evaluates how the duration of the LGCP exploration phase influences downstream SAC performance. All configurations maintain similar performance during their respective LGCP phases, with rewards stabilising around 380. The critical differences emerge during and after the policy transition. The 10-episode configuration exhibits a pronounced performance dip as it shifts to SAC, dropping approximately 12% before recovering. This degradation occurs because the shorter warm-start phase collects

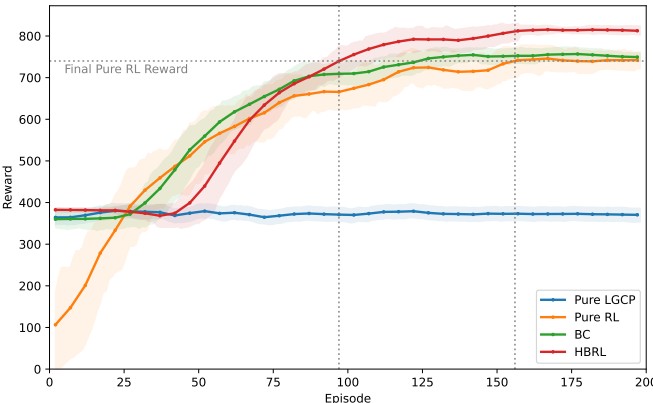

**Figure 4:** Reward comparison between Pure LGCP, Pure RL, Behavior Cloning and HBRL frameworks.

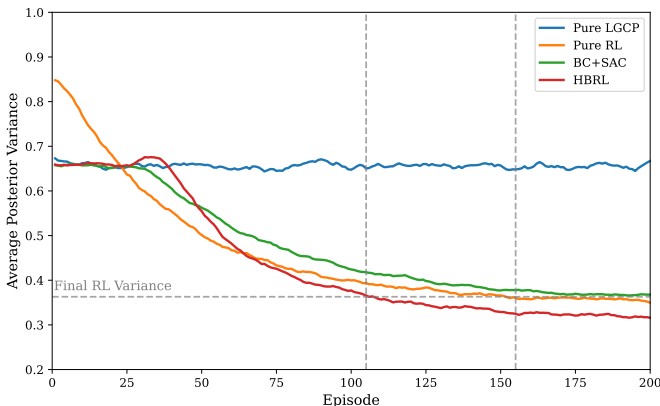

**Figure 5:** Posterior Variance comparison between Pure LGCP, Pure RL and HBRL. Lower values indicate higher confidence in the inferred demand field.

only 2000 replay buffer transitions, leaving the SAC agent with limited expert demonstrations to learn from. In contrast, longer warm-start durations accumulate more diverse state-action-reward transitions covering broader regions of the service area, enabling more effective policy initialisation.

A short warm-start of 10 episodes also yields the lowest final reward, as the replay buffer contains limited behavioural structure. Increasing the warm-start duration increases the final reward, with 20 and 30 episode warm-up yielding a clear improvement in rewards of approximately 6% and 5.2%, respectively. This occurs because longer LGCP exploration populates the replay buffer with a more diverse set of informative trajectories, enabling SAC to begin learning from substantially better initial conditions. However, the final reward of 30 episode vs 50 episode warm-start is identical. This is most likely attributable to excessive warm-starting which biases the replay buffer heavily towards LGCP-style behaviour, making SAC slower to adapt its own policy and resulting in a lagging training graph.

The LGCP PathMI horizon ablation in Figure 8 demonstrates that planning depth strongly influences final reward. The myopic setting (L=1) achieves a substantially low median reward as compared to non-myopic configurations, as single-step planning greedily selects actions based on immediate information gain without considering future observational opportunities. Nevertheless, even this purely local strategy outperforms the pure RL baseline (740) by approximately 4.5%. Increasing the horizon to L=3 and L=5 yields an improvement of 4.8% and 6.1%, respectively. These results indicate that moderate non-myopic planning enables UAVs to anticipate future information gain and commit to globally beneficial trajectories. However, extending beyond L=5 produces diminishing returns. L=7 maintains comparable performance while adding to the computational cost. L=9 degrades slightly, as uncertainty in long-horizon predictions reduces their reliability.

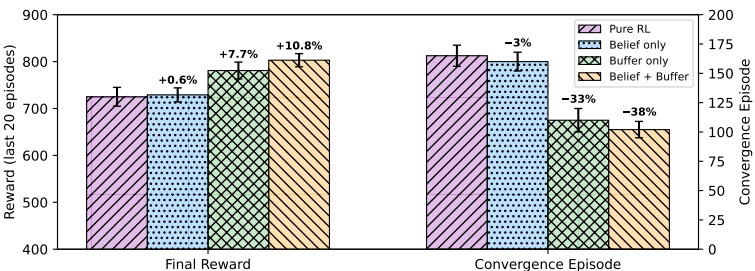

**Figure 6:** Comparison of reward and episodes to reach Pure RL reward for all three transfer channel scenarios.

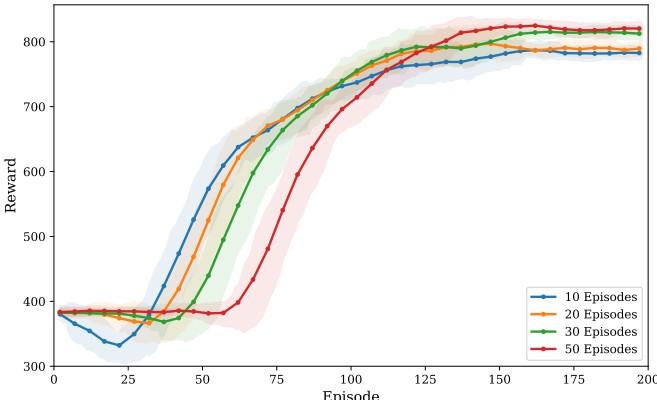

**Figure 7:** Effect of LGCP warm-start duration on SAC training. Different warm-start lengths determine the transition point from LGCP exploration (Phase-1) to SAC optimization(Phase-2).

### 5.4 Scalability and Coordination Analysis

Figure 9 evaluates the scalability of HBRL across multi-UAV configurations. The main plot demonstrates that all configurations successfully learn coordinated coverage policies, with final rewards of 820, 960, and 1080 for 2, 3, and 4 UAVs, respectively. The upper-left inset highlights the transition dip phenomenon occurring during episodes 30-50, where performance temporarily degrades as the SAC policy takes over from PathMI-guided exploration. It can be seen that this degradation is more pronounced for larger fleets, reflecting the increased complexity of learning coordinated behaviors when more agents must avoid spatial overlap. The lower-right inset quantifies scaling efficiency by comparing observed total reward against ideal linear scaling. The sublinear returns can be attributable to increasing coordination overhead, which is captured by the variance-normalized overlap penalty that discourages redundant coverage of already well-characterized regions.

Figure 10 compares final performance under three levels of operational-region overlap and different coordination strategies. When the operational regions do not overlap, all methods achieve similar rewards, and both fixed and variance-normalized penalties have negligible impact as UAV sensing footprints rarely overlap. In partial overlap, coordination becomes beneficial: the fixed penalty yields improvements by discouraging some redundant sensing, while the proposed variance-normalized penalty performs best, improving the final reward by approximately 6%. This gain arises from selectively allowing overlap in high-uncertainty regions while penalizing redundancy elsewhere. The benefit of adaptive coordination is most pronounced in the high-overlap regime, where substantial overlap between operational regions leads to frequent sensing conflicts and significant performance degradation in the absence of coordination. In this case, the variance-normalized

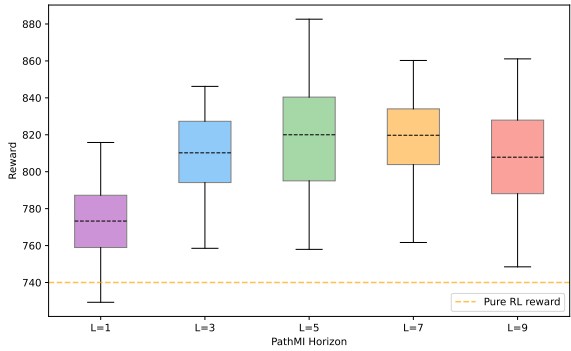

**Figure 8:** Effect of PathMI planning horizon on final reward.

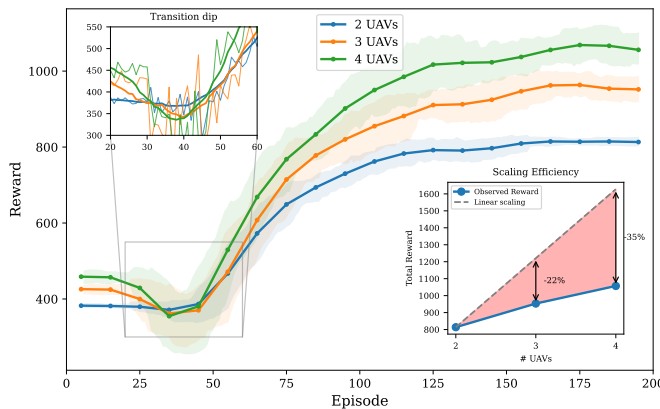

**Figure 9:** Learning performance comparison under varying numbers of UAVs. Increasing the number of agents improves overall reward but exhibits sub-linear scaling due to coordination overhead and redundant coverage

penalty improves final reward by about 34% relative to no penalty, and by more than 14% compared to the fixed penalty.

## 5.5 Belief Dynamics and Uncertainty Calibration

Figure 11 examines the role of temporal belief decay in maintaining accurate spatial awareness. Temporal decay increases posterior variance for cells not recently visited, prompting revisiting previously visited areas. Enabling temporal decay achieves 8.2% higher final reward, demonstrating the value of maintaining fresh beliefs. However, it reveals a key insight: with temporal decay turned on, the system reports higher posterior variance because it correctly acknowledges uncertainty growth in unvisited regions. This elevated variance drives the planner to revisit areas with stale information, keeping beliefs calibrated with updated demand patterns. Conversely, with no temporal decay of information, variance remains artificially low as the system falsely assumes past observations remain valid indefinitely. This leads to no revisitation despite information staleness, resulting in decisions based on outdated beliefs and, consequently, lower cumulative reward.

Figure 12 presents the sensitivity analysis for reward weights $(\omega_1, \omega_2, \omega_3)$. The exploration weight $\omega_2$ exhibits a broad optimal plateau between 0.4 and 0.6, with $\omega_2 = 0$ causing 11% degradation due to insufficient exploration incentive. Similarly, while extreme $\omega_3$ values degrade performance, the optimal region spans a broad range, which indicates robustness to moderate tuning variations. The service weight $\omega_1$ shows best and comparable reward for $\omega_1 = 4, 5$, with $\omega_1 = 6$ performing worse, indicating that over-emphasis on immediate service rewards causes the agent to exploit known hotspots at the expense of discovering new demand regions.

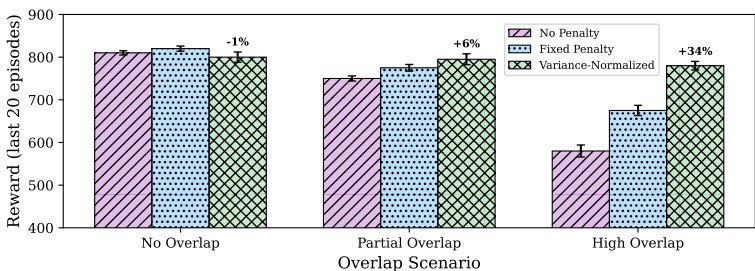

**Figure 10:** Comparison of reward under various overlap penalty scenarios.

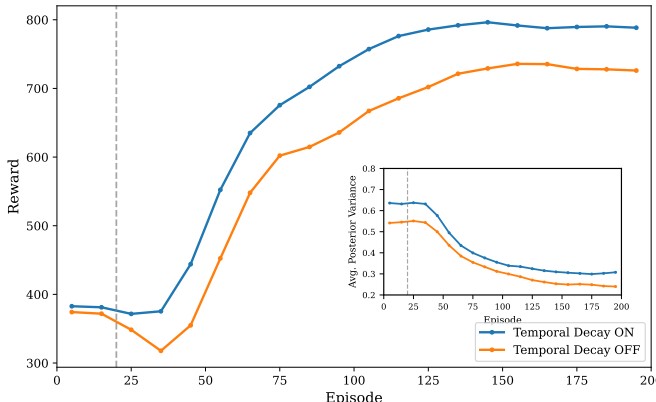

**Figure 11:** Impact of temporal decay on HBRL performance: (a) reward convergence and (b) belief uncertainty evolution. The dashed line denotes the warm-start transition point.

## 5.6 Robustness to Experience Loss

Figure 13 evaluates robustness under stochastic replay corruption. We replace deterministic replay insertion with independent Bernoulli thinning, where each transition $(\mathbf{s}_t, \mathbf{a}_t, r_t, \mathbf{s}_{t+1})$ is inserted into the buffer with probability $1 - p_{\text{loss}}$, for $p_{\text{loss}} \in \{0.0, 0.2, 0.4, 0.6, 0.8\}$. All results are averaged over five random seeds, paired across all methods and $p_{\text{loss}}$ values. Performance degrades smoothly as $p_{\text{loss}}$ increases, indicating graceful degradation under reduced replay availability. At moderate loss levels ($p_{\text{loss}} \leq 0.4$), both methods maintain stable learning dynamics, while higher loss primarily slows convergence due to delayed buffer saturation. The inset plots show that increasing $p_{\text{loss}}$ reduces effective buffer fill-rate, thereby limiting the diversity of stored transitions. Figure 14 summarizes final performance over the last 20 episodes. Across all loss levels, degradation is gradual rather than catastrophic, demonstrating the robustness of the proposed framework to experience/replay loss. Convergence here is defined as the first episode reaching 95% of each configuration's *own* final reward.

## 6 Conclusion

This paper presents a framework for coordinating multiple mobile agents under spatial uncertainty. The work is motivated by the challenge of coordinated learning under spatial uncertainty, where agents must operate with incomplete and evolving information. We propose HBRL, a belief-guided warm-start reinforcement learning framework, that integrates LGCP-based spatial inference with non-myopic information-driven planning and off-policy deep reinforcement learning. The approach enables coordinated policy learning while accounting for uncertainty in spatial demand. Performance was evaluated by comparison with planning-only

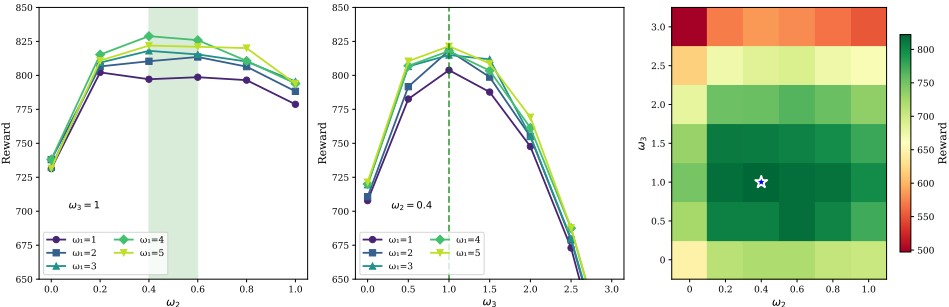

**Figure 12:** Reward weight sensitivity analysis. (a) Effect of exploration weight $\omega_2$ on final reward with coordination weight fixed at $\omega_3 = 1.0$. The shaded region indicates the optimal range $\omega_2 \in [0.4, 0.6]$. (b) Effect of coordination weight $\omega_3$ on final reward with exploration weight fixed at $\omega_2 = 0.4$. (c) Reward heatmap over the $(\omega_2, \omega_3)$ configuration space with $\omega_1 = 5$. The star indicates the default configuration.

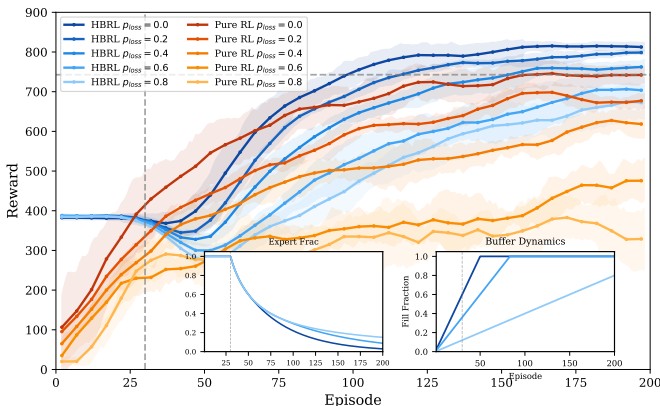

**Figure 13:** Training curves for different $p_{\text{loss}}$ values

and learning-only baselines across different overlap configurations, workload intensities, and training settings. The results show that the HBRL achieves faster convergence and consistently higher long-term reward, with significant reward and convergence speed improvements compared to baseline approaches. The framework is well-suited to multiple mobile agent settings where shared spatial structure can be exploited to accelerate learning.

However, this work is restricted to the initial warm-start phase, which suggests extending the framework toward continuous belief–policy co-adaptation as a direction for future work. Additionally, the impact of scaling to significantly larger fleet sizes merits further investigation.

The simulated reward is a proxy composed of service, exploration, and coordination terms weighted by $(\omega_1, \omega_2, \omega_3)$. Its correspondence to application-specific deployment utility is a modelling assumption that warrants validation in relevant deployment scenarios. Relatedly, sim-to-real transferability, the bridging the gap between simulated policies and behaviour on real hardware, remains an open direction for this framework; hardware-in-the-loop evaluation is a natural next step.

While the present empirical evaluation is on wireless service provisioning, the core framework components are domain-agnostic: LGCP posterior inference, PathMI planning, dual-channel knowledge transfer, and the variance-normalized overlap penalty may apply to any spatially heterogeneous multi-agent exploration setting. Only the observation model and sensing-footprint geometry are task-specific and require adaptation when moving to a new domain. Applying the framework to other spatial exploration contexts such as environmental monitoring or precision agriculture remains an open direction.

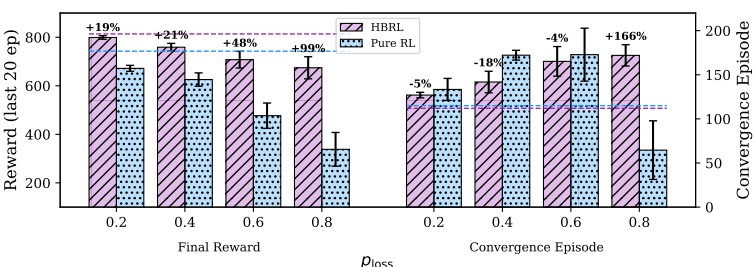

**Figure 14:** Final Reward and Convergence vs. $p_{\text{loss}}$

## 7 Acknowledgment

Acknowledgements removed for double-blind review. The authors acknowledge use of generative AI tools (OpenAI ChatGPT and Grammarly) to assist with language refinement. All conceptual development, experimental design, analysis, and conclusions were conducted and verified by the authors.

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

# A  Warm Start Exploration Strategies

To motivate the choice of PathMI planning for Phase 1, we compare four action-selection methods in a planning-only setting (without RL). All strategies share the same belief model and differ only in trajectory planning: **PathMI** (multi-step lookahead via (30)), **UCB** (greedy, $\mu_c + \kappa\sqrt{\sigma_c^2}$, $\kappa=2$), **Lawnmower** (boustrophedon sweep), and **Random** (uniform direction). To isolate the planning strategy from other framework components, we evaluate on a $1000 \times 1000$ m scenario ($\Delta=10$ m, two Gaussian hotspots, $N=2$ UAVs, $r_c=100$ m and 10 seeds per configuration).

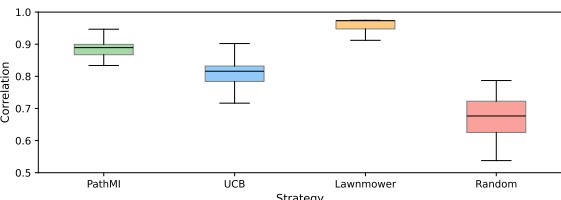

**Figure 15:** Strategy vs. Correlation

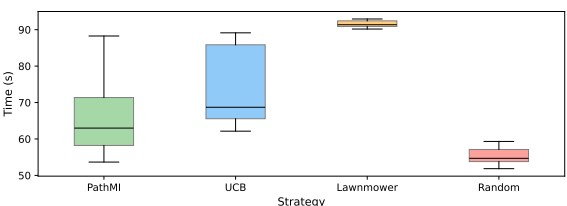

**Figure 16:** Strategy vs. Runtime

Figures 15 and 16 report Pearson correlation between estimated and ground-truth intensity and per-episode runtime. PathMI achieves median correlation 0.89, approaching Lawnmower (0.96) while requiring around 30% less time. UCB attains 0.81 with higher variance due to greedy local optima. Lawnmower achieves the highest correlation but cannot adapt to observed demand. These results support PathMI as the Phase 1 strategy, offering the best accuracy–computation trade-off among adaptive methods.

# B  Learned Belief Intensity

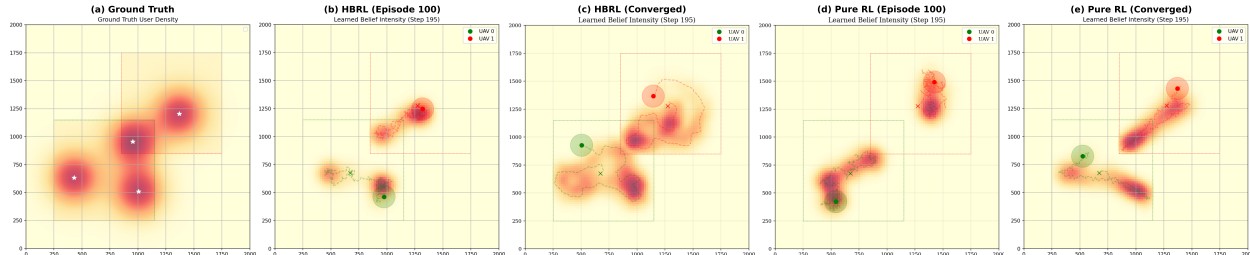

**Figure 17:** Learned belief intensity maps at episode 100 and convergence for HBRL and Pure RL, compared against the ground truth

Figure 17 presents the final belief intensity maps reconstructed by each method at selected training episodes. At episode 100, HBRL has already identified all four density centers across both operational regions, producing a belief map closely resembling the ground truth. In contrast, Pure RL at episode 100 has only discovered one density center per region, reflecting its slower learning and confirming that the performance gap due to sample efficiency. Animated visualizations of the belief evolution over the course of each episode are provided as supplementary material.

