# OpenReview forum: "Hybrid Belief–Reinforcement Learning for Efficient Coordinated Spatial Exploration"
_TMLR — Rejected by TMLR_

### Review · Reviewer_d9th · 2026-04-13

**Summary Of Contributions:**

The manuscript deals with multi-agent systems.
The authors claim that they are presenting a new approach, which is “a belief-guided warm-start reinforcement learning framework that integrates LGCP-based spatial inference with non-myopic information-driven planning and off-policy deep reinforcement learning.”

**Strengths**

* The text appears to be grammatically correct.

**Weaknesses**

* The text is incomprehensible. Key terms are not sufficiently clarified.
* Claims are made that are not sufficiently substantiated.
* The integration into existing literature is insufficient.
* Limitations and preconditions are not sufficiently clarified.
* The baselines used are insufficient.

**Additional Comments:**

Terms are used that are not explained until later, either through a reference or a definition provided in the text.

Abbreviations are used that are not defined.

**Audience:**

No

**Audience Explanation:**

Since the text is largely incomprehensible, many of its statements appear to make no sense, and the manuscript lacks the necessary scientific rigour, no meaningful insights can be gained from it.

**Claims And Evidence:**

No

**Claims Explanation:**

First, it should be noted that many statements are unclear and do not seem to make sense.

The text is largely vague. Key terms are not sufficiently defined.

For example, the following terms are not sufficiently defined:
* behavioral transfer
* dual-channel knowledge transfer
* belief state transfer
* behavioral knowledge transfer
* spatial demand information


In short, the manuscript lacks the necessary scientific rigour.

**Requested Changes:**

I see no way to turn this manuscript into a paper worthy of publication.

My suggestion is to try to
* break it down into several shorter papers.
* Step by step, one idea per paper.
* Clearly define each idea,
* thoroughly discuss the literature,
* use strong baselines, and
* clearly explain the limitations and assumptions.

---

> ### Author Response · Authors · 2026-04-17
> **Response to Comments 1-3**
>
> Thank you for the review of our manuscript. We appreciate the feedback and have made relevant clarifications and improvements to enhance the quality of the manuscript. All changes in response to Reviewer d9th comments are highlighted in **yellow** in the revised manuscript.
>
> ## Comment 1. Key terms not sufficiently defined
>
> We thank the reviewer for identifying these specific terms. Each is addressed below:
>
> - ***Belief state transfer.*** Defined on first use in the Contributions (Section 1.1) as "which provides the RL agent with an informed prior for early policy updates". The underlying concept "belief state" is a standard term from the POMDP literature; a citation to Kaelbling et al. (1998) has been added at its first mathematical introduction in Section 3.6.
> - ***Behavioral transfer.*** Defined on first use in the Contributions (Section 1.1) as "which seeds the replay buffer with LGCP-generated exploration trajectories". Formally specified in Section 4.3.2 via Equations 34–35.
> - ***Dual-channel knowledge transfer.*** Defined on first use in the Abstract as the combination of "belief state transfer" and "replay buffer seeding". Formally introduced in Section 4.3 as the umbrella mechanism for the two channels above.
> - ***Behavioral knowledge transfer.*** In the original submission this appeared as the title of Section 4.3.2 and in the Figure 2 caption as a variant of "behavioral transfer". In the revision the terminology is unified to "behavioral transfer" throughout to remove any ambiguity.
> - ***Spatial demand information.*** A descriptive phrase (not a coined technical term) that refers to the learned representation of the underlying *spatial demand*, formally defined in Section 3.3 via Equations 4–5. The distinction between ground-truth demand and the agents' learned belief is made explicit in Remark 1.
>
> Additionally, a one-sentence introduction to both concepts, the belief state and learning from behavioural demonstrations, has been added to the Introduction (Section 1) with supporting citations, so that readers encounter these ideas before the Contributions list.
>
> We welcome the opportunity to make any additional clarifications that may be suggested.
>
> ## Comment 2. Claims not sufficiently substantiated
>
> Each quantitative claim in the manuscript is experimentally established and supported by a dedicated figure or ablation:
>
> - **10.8% higher cumulative reward**: Figure 4 (Section 5.2).
> - **38% faster convergence**: Figures 4 and 6 (Section 5.2).
> - **Dual-channel outperforms single-channel transfer**: Figure 6 ablation (Section 5.2): $+0.6\%$ (belief only), $+7.7\%$ (buffer only), $+10.8\%$ (both).
> - **Non-myopic planning benefit**: Figure 8, horizon ablation $L \in \{1,3,5,7,9\}$ (Section 5.3).
> - **Scalability across fleet size**: Figure 9, $N \in \{2,3,4\}$ UAVs (Section 5.4).
> - **Variance-normalized overlap penalty**: Figure 10, three overlap regimes (Section 5.4).
> - **Temporal belief decay**: Figure 11, $+8.2\%$ reward (Section 5.5).
> - **Robustness to replay loss**: Figures 13–14, $p_\text{loss} \in \{0.0, 0.2, 0.4, 0.6, 0.8\}$ (Section 5.6).
>
> We welcome the opportunity to provide additional substantiation for any specific claim(s) considered to be insufficiently substantiated.
>
> ## Comment 3. Literature integration insufficient
>
> The manuscript contains a comprehensive literature integration:
>
> - **Section 2 (Related Work and Motivation)** comprises six subsections: Informative Path Planning (2.1), Spatial Statistical Modelling (2.2), RL-Based Coordination (2.3), Transfer Learning and Warm-Starting (2.4), Trajectory Optimization (2.5), and Research Gaps and Motivation (2.6). We believe that Section 2.6 explicitly integrates our contribution with existing literature.
> - **Table 1** provides direct positioning of the proposed framework with regard to eight prior methods spanning five relevant integrative research aspects.
>
> We welcome the opportunity to consider and potentially integrate any additional relevant literature we may have omitted to include in our analysis.

---

> ### Author Response · Authors · 2026-04-17
> **Response to Comments 4-7**
>
> ## Comment 4. Limitations not sufficiently clarified
>
> Limitations, assumptions, and preconditions are stated at the following manuscript locations:
>
> - **Remark 2** (Section 3.4): Poisson approximation assumption requires $\pi_\text{req} \leq 0.05$. This is justified by classical Poisson limit results.
> - **Remark 3** (Section 3.5.3): Diagonal Laplace approximation does not yield calibrated marginal variances. This is used as a relative uncertainty surrogate.
> - **Remark 4** (Section 3.7.3): Adaptive behaviour and limits of the variance-normalized overlap penalty.
> - **Remark 7** (Section 4.4.1): Compressed belief summary deliberately sacrifices local spatial detail in favour of state-dimension scalability.
> - **Section 3.1**: Assumption of predefined operational subregions $\mathcal{A}_i \subset \mathcal{A}$ and grid discretisation.
> - **Section 4.6.3**: Parameter-sharing for very large fleet sizes is identified as an interesting avenue for future work.
> - **Section 6 (Conclusion)**: Explicitly identifies: (i) restriction to the initial warm-start phase, (ii) impact of significantly larger fleet sizes, and (iii) applicability to non-wireless domains, as relevant areas for future work.
>
> We welcome the identification of any other limitations or preconditions that may require further clarification, and will be happy to revise accordingly.
>
> ## Comment 5. Baselines insufficient
>
> Our methodology follows a comprehensive ablation approach. Eight baselines are reported across Section 5 and Appendix A:
>
> - **Learning-based baselines (Section 5.1):** Pure LGCP-PathMI (planning only, no RL); Pure SAC (cold-start RL, no warm-start); SAC + Belief Transfer Only (Channel 1 ablation); SAC + Buffer Seeding Only (Channel 2 ablation); BC + SAC (Behavior Cloning followed by SAC fine-tuning).
> - **Planning-strategy baselines (Appendix A):** UCB (greedy upper-confidence-bound); Lawnmower (systematic boustrophedon sweep); Random (uniform-direction sampling).
>
> We welcome the reviewer's suggestion of any additional baseline they consider important and will include it where feasible.
>
> ## Comment 6. Suggestion to split into multiple papers
>
> This paper presents a single central idea: **Hybrid Belief–Reinforcement Learning (HBRL)**. This is a framework that sequentially combines model-based Bayesian spatial inference (LGCP + PathMI) with model-free reinforcement learning (SAC). We believe that the two phases and the dual-channel knowledge transfer between them are not separable ideas that can stand independently in shorter papers given that they are the constituent elements of one novel HBRL framework.
>
> Regarding the specific elements of the reviewer's suggestion:
>
> - **Clearly define each idea.** Each key term is defined on first use in the Abstract and Contributions. Please refer to our response to Comment 1, above.
> - **Thoroughly discuss the literature.** Section 2 comprises six subsections, where Table 1 compares against eight relevant prior methods and situates the manuscript's contributions accordingly, where more details are provided in our response to Comment 3, above.
> - **Use strong baselines.** We refer to our response to Comment 5, above.
> - **Clearly explain the limitations and assumptions.** Remarks 2, 3, 4, and 7, and in Sections 3.1, 3.3, 4.6.3, and 6 establish and explain our limitations and assumptions, where more detail is provided in our response to Comment 4, above.
>
> ## Comment 7. Abbreviations not defined
>
> We thank the reviewer for highlighting this and the opportunity to enhance the quality of our manuscript. We have identified and justified on first use the following abbreviations: UAV, GP, MEC, DQN, NOMA, MARL, MAP, MDP, PPO, and DDPG. We would be happy to include a glossary of terms and/or acronyms as an appendix in the next revision should this be deemed editorially desirable.
>
> ## Summary of revisions
>
> 1. Abbreviations expanded at first use, including: UAV, GP, MEC, DQN, NOMA, MARL, MAP, MDP, PPO, DDPG.
> 2. Nomenclature unified: "belief state transfer" (Channel 1) and "behavioral transfer" (Channel 2) used consistently across the Abstract, Contributions, architecture description, Figure 2 caption, and Section 4.3.
> 3. One-sentence introduction to both key concepts (belief state and learning from behavioural demonstrations) added to the Introduction with supporting citations (Kaelbling et al., 1998).
> 4. Citation to Kaelbling et al. (1998) added at the first mathematical introduction of the belief state in Section 3.6.
> 5. Enhanced signposting paragraph at the end of Section 1 to guide readers through the structure of the paper.
>
> All changes are highlighted in **yellow** in the revised manuscript. Original text proposed to be deleted remains in the revision using strikethrough.

---

> > ### Comment · Reviewer_d9th · 2026-04-30
> > **Comments on the Revised PDF of 28 Apr 2026**
> >
> > The changes are a step in the right direction, but their scope is nowhere near sufficient. The fundamental problems remain unchanged. In my opinion, addressing these issues requires extremely extensive changes, amounting to a complete redesign and a complete rewrite. It is necessary to define the terms used right at the beginning; otherwise, it is not sufficiently clear what is meant.
> >
> > Furthermore, the statement “The framework is well-suited to multiple mobile agent settings where shared spatial structure can be exploited to accelerate learning” is an unacceptable exaggeration, since only N=2 was investigated.

---

### Review · Reviewer_QReF · 2026-04-15

**Summary Of Contributions:**

This paper studies the coordination of multiple atuonomous agents in exploring and serving spatially heterogeneous demand. It presents a hybrid belief-reinforcement learning (HBRL) framework to combine model-based approaches and deep reinfocement learning. The framework involves two phase. In the first phase, agents construct spatial belifes using a Log-Gaussian Cox Process (LGCP) and execute information-driven trajectories guided by a Pathwise Mutual Information (PathMI) pllaner with multi-step lookahead. In the second phase, the framework applies a Soft Actor-Critic (SAC) agent and applies deep reinforcement learning with three reward terms. The framework is evaluated on a multi-UAV wireless service provisioning task using simulated data.

**Additional Comments:**

I have no experience in the field of UAV and my judgement above represents the opinion of general ML audience.

**Audience:**

Yes

**Audience Explanation:**

The paper introduces a new framework for multi-agent coordination problem. This might be interesting to people in the field especially to people who are working on UAV. However, I have low confidence in my judgment here given I have no experience with UAV and am not familar with the relevant literature.

**Broader Impact Concerns:**

I do not see potential ethical concerns in this work.

**Claims And Evidence:**

No

**Claims Explanation:**

I am concerned about the main argument of the paper.
1. The paper proposes a new learning framework for multi-agent coordination in exploring and serving spatially heterogenous demand. However, the experiment is only conducted in the multi-UAV wireless service provisioning task. Given that the framework is complicated and involves multiple hyperparameters, it is unclear whether it will generalize to other multi-agent coordination scenarios.
2. Also, for the multi-UAV wireless service provisioning experiment, the paper only tests $N=2$. Since the paper argues the difference between multi-UAV scenario and single-UAV scenario, I think $N$ is a critical factor in the problem complexity and it is unclear whether the proposed framework will still yield better performance when we increase $N$.
3. The main comparison with existing work uses the RL reward in the simulated setup. it's unclear whether the reward completely reflects the real-world utility given that this scenario itself involves multiple reward terms and there is a common challenge of sim-to-real transferability in deep RL.

**Requested Changes:**

1. Explain how this framework can be used for other multi-agent coordination scenarios besides multi-UAV wireless service provisioning task and conduct corresponding experiments to demonstrate its effectiveness. If conducting experiments in other setup is hard, I would suggest describing the contribution as introducing a framework for multi-UAV wireless service provisioning task.
2. Add experiments where $N$ is greater than 2.
3. Deploy the trained policy on real UAVs to verify that the policy is actually effective in real setup.

---

> ### Author Response · Authors · 2026-04-17
> **Response to Comments 1 and 2**
>
> Thank you for the careful and constructive review of our manuscript. We appreciate the detailed feedback and have made relevant clarifications and improvements. All changes in response to Reviewer QReF comments are highlighted in **light blue** in the revised manuscript.
>
> ## Comment 1. Generalization beyond multi-UAV
>
> The contribution of this work is a general learning framework for multi‑agent coordination under spatial uncertainty, rather than a UAV‑specific system. HBRL addresses a canonical ML problem: how to combine structured uncertainty estimation with sample‑efficient policy learning when agents must act under partial observability. The multi‑UAV wireless service provisioning scenario is used as an exemplar instantiation to make the framework concrete, and where the same learning architecture applies to any setting with mobile agents interacting with unknown, spatially distributed event processes such as mapping of pollution hotspots, water quality, disease outbreaks, precision agriculture, etc..
>
> To maintain focus on the learning and inference mechanisms, we intentionally abstract away from application-specific engineering details such as wireless channel models, robot dynamics or UAV-specific energy models. The models we have used, i.e., point-mass agent dynamics (Equation 1), circular sensing footprints (Equation 3), and binomial event observations (Equation 6), are minimal and standard across mobile sensing, exploration and coverage tasks in the literature.
>
> The manuscript already addresses domain generalisation in two places:
>
> - **Introduction (page 1).** The motivating applications are listed as "environmental monitoring, dynamic wireless connectivity scaling, precision agriculture, disaster response, and infrastructure inspection".
> - **Section 6 (Conclusion).** "While the framework is evaluated on wireless service provisioning, the underlying LGCP–SAC structure is domain-agnostic, and applying it to other spatial exploration contexts such as environmental monitoring or precision agriculture remains an open direction."
>
> In the revision, we have further strengthened this framing as follows:
>
> - We clarify the final Contributions bullet to state: "We instantiate and evaluate the proposed framework using multi-UAV wireless service provisioning as an **exemplar scenario**."
> - We expanded Section 6 (Conclusion) to detail which framework components are domain-agnostic (LGCP posterior inference, PathMI planning, dual-channel knowledge transfer, variance-normalized overlap penalty) and which are task-specific (observation model and sensing geometry).
>
> We hope this addresses the concern regarding generality beyond the multi‑UAV setting.
>
> ## Comment 2. Fleet size $N$ experiments
>
> We agree that fleet size $N$ is central to problem complexity. Experiments across multiple values of $N$ are reported in the manuscript:
>
> - **Section 5.4** (*Scalability and Coordination Analysis*) and **Figure 9** evaluate the framework with $N \in \{2, 3, 4\}$ UAVs.
> - Figure 9 reports final rewards of 820, 960, and 1080 for $N = 2, 3, 4$ respectively, together with a scaling-efficiency analysis (sub-linear scaling due to coordination overhead and redundant coverage) and the transition-dip phenomenon during Phase 2 onset, which is more pronounced for larger fleets.
>
> The complexity analysis in Section 4.6 explicitly analyses the dependence on $N$: the state dimension grows as $3N+4$ and the action dimension as $2N$. For very large $N$, the joint action space becomes prohibitively large for a single shared policy, necessitating parameter-sharing or hierarchical techniques — identified in Section 4.6.3 as an open challenge for future work.
>
> If the reviewer would like to see additional data points at larger $N$ (e.g., $N = 5, 6$), we are happy to include them in the revision.

---

> ### Author Response · Authors · 2026-04-17
> **Response to Comments 3-5**
>
> ## Comment 3. Reward as proxy for real-world utility
>
> We appreciate this observation. We address the reward–utility mapping as follows:
>
> - **Reward-weight robustness.** Section 5.5 and Figure 12 present a sensitivity analysis over the reward weights $(\omega_1, \omega_2, \omega_3)$, showing a broad optimal plateau ($\omega_2 \in [0.4, 0.6]$) and robustness to moderate tuning variations. This demonstrates that the reported performance does not rely on narrow parameter tuning.
> - **Operational interpretation.** In the revision, a new paragraph has been added to Section 4.4.3 (Reward Function) mapping each reward term to established operational objectives with supporting citations: $R_\text{service}$ (served demand) aligns with throughput optimisation (Wu et al., 2018) and user quality of service (Hao et al., 2024); $R_\text{explore}$ (belief improvement) aligns with demand-uncertainty reduction (Krause et al., 2008); $C_\text{coord}$ aligns with spectrum-contention avoidance and UAV energy budget (Zeng et al., 2019).
> - **Limitation acknowledged.** We agree that the correspondence between simulated reward and application-specific deployment utility is a modelling assumption. This is now explicitly stated as a limitation in Section 6.
>
> The specific mapping between reward terms and deployment utility is open to study in any relevant scenario; the present work focuses on demonstrating the framework-level contribution.
>
> ## Comment 4. Real UAV deployment
>
> The nature of the HBRL contribution is a framework for coordinating multiple mobile agents under spatial uncertainty, compared against state-of-the-art baselines in a controlled setting. We have leveraged models that we believe are appropriate for demonstrating the intellectual contribution, intentionally stopping short of high-fidelity application-specific simulations (e.g., wireless channel models, robot-specific propulsion energy, dynamics, etc.). Real-world deployment would require substantial engineering work (flight certification, airspace authorisation, safety systems, channel modelling, energy management, etc.) that is beyond the scope of this work. Nonetheless, we believe the framework is well-suited for emergent 6G-era autonomous service provisioning contexts and this is why we chose the illustrative scenario.
>
> We note that the manuscript does provide evidence of robustness that will be relevant to real deployments:
>
> - **Section 5.6 and Figures 13–14** evaluate robustness under stochastic replay-buffer loss ($p_\text{loss} \in \{0, 0.2, 0.4, 0.6, 0.8\}$), modelled as independent Bernoulli thinning of stored transitions. Such probabilistic data loss is representative of conditions that would arise in real deployments (e.g., communication dropouts, sensor failures, etc.). The framework is demonstrated to degrade gracefully across all loss levels rather than failing catastrophically, evidencing robustness to the kinds of perturbation expected in practice.
>
> Sim-to-real transferability and hardware-in-the-loop evaluation are now explicitly listed as future-work directions in Section 6.
>
> ## Comment 5. Reviewer's note on UAV expertise
>
> We appreciate the reviewer's candour. We note again that the framework is not UAV-specific and may apply to numerous multi-agent coordination scenarios with unknown spatial demand. We hope that the clarifications above, particularly the domain-agnostic framing in Comment 1 and the operational interpretation in Comment 3, sufficiently address the concerns from a general ML perspective and particularly for readers interested in uncertainty-aware reinforcement learning and multi-agent coordination.
>
> ## Summary of revisions
>
> 1. Final Contributions bullet revised to state "exemplar scenario" framing for multi-UAV wireless service provisioning.
> 2. Opening sentence added to Section 6 (Conclusion) framing the paper's contribution as coordinating multiple mobile agents under spatial uncertainty.
> 3. Expanded generalisation discussion in Section 6: domain-agnostic vs. task-specific components identified.
> 4. Operational interpretation of reward terms added to Section 4.4.3 (Reward Function) with citations mapping each term to established objectives (throughput, QoS, uncertainty reduction, interference, energy).
> 5. Deliberate abstraction from application-specific models noted in Section 4.4.3, citing Rizvi & Boyle (2025).
> 6. Reward-proxy and sim-to-real limitations explicitly listed in Section 6.
>
> All changes are highlighted in **light blue** in the revised manuscript.

---

### Review · Reviewer_FDQV · 2026-04-23

**Summary Of Contributions:**

This work considers a hybrid approach for multi-agent spatial exploration when demands and planning trajectories are unknown. In particular, the authors combine deep reinforcement learning with spatial belief modeling in order to both encourage accurate policy learning with uncertainty quantification. The method is divided into two stages, where in the first stage a Log-Gaussian Cox Process is used to construct spatial beliefs with trajectories generated by a PathMI planner. Then an SAC agent is learned with prior exploration trajectories used as warm starts.

**Strengths:**
- The problem is important and motivated well.
- The experimental results are extensive and several ablations help elucidate the benefits of each component. For example, the variance-normalized overlap penalty is intuitive and experiments show that this term is indeed useful in situations where there is high overlap (e.g., Fig 10).


**Weaknesses:**
- In some places, the proposed form of specific criteria can feel heuristic and could benefit from additional justification/discussion, especially for researchers outside of the area. This is discussed more below.

**Audience:**

Yes

**Audience Explanation:**

Yes, I believe researchers in reinforcement learning, motion planning, and spatial exploration will find this work interesting.

**Claims And Evidence:**

Yes

**Claims Explanation:**

The empirical validation of the approach is rigorous and extensive.

**Requested Changes:**

The authors can consider the following questions when revising their work. Addressing such questions would simply strengthen the work in my view:
- How sensitive are the results to the use of the diagonal Laplace proxy? It could be interesting to see some low-dimensional examples in which the full posterior uncertainty is tractable, and then compare whether the action rankings agree with these fully-calibrated uncertainty estimates.
- For the PathMI objective, is there a way to derive this from an underlying information-theoretic quantity? A derivation of this (even in the appendix) could help give a more rigorous justification for the specific form considered here along with certain choices, such as $\xi_c = 1/(1+n_c)$.

---

> ### Author Response · Authors · 2026-04-28
>
> We thank the reviewer for the careful and constructive reading and for two targeted suggestions. Both suggestions have prompted small edits to the manuscript, highlighted in **light green**, as detailed below.
>
> ---
>
> **Comment 1.** *"How sensitive are the results to the use of the diagonal Laplace proxy? It could be interesting to see some low-dimensional examples in which the full posterior uncertainty is tractable, and then compare whether the action rankings agree with these fully-calibrated uncertainty estimates."*
>
> **Response.** A direct comparison between the diagonal proxy $1/[\mathbf{H}]\_{cc}$ and the full marginal variance $[\mathbf{H}^{-1}]\_{cc}$ on low-dimensional grids is an interesting way to verify that the proxy carries the rank information on which exploration decisions depend. Accordingly, we have added a one-sentence acknowledgment to Section 3.5.3 noting that such comparison is tractable only at small grid scales, and that the proxy is adopted as it preserves the relative uncertainty ordering on which exploration decisions depend. We have not extended this comparison to the operational grid size used in this work because direct inversion of $\mathbf{H}$ scales cubically in $N$ and is not feasible at that scale. The results of the suggested comparison, which we carried out on grids small enough for direct inversion using the main parameters ($\tau=1.0$, $\beta=0.2$), are reported in the table below.
>
> | Grid | $N$ | Spearman $\rho$ | PLA ($L=3$) | PLA ($L=5$) | $t_\text{full}/t_\text{diag}$ |
> |---|---|---|---|---|---|
> | $5\times 5$ | $25$ | $0.985 \pm 0.009$ | $0.990$ | $0.990$ | $\sim 60\times$ |
> | $10\times 10$ | $100$ | $0.979 \pm 0.008$ | $0.990$ | $0.985$ | $\sim 90\times$ |
> | $15\times 15$ | $225$ | $0.971 \pm 0.007$ | $0.985$ | $0.975$ | $\sim 290\times$ |
> | $20\times 20$ | $400$ | $0.964 \pm 0.006$ | $0.993$ | $0.993$ | $\sim 510\times$ |
> | $30\times 30$ | $900$ | $0.957 \pm 0.004$ | $0.990$ | $0.990$ | $\sim 1300\times$ |
>
> *Agreement between the diagonal proxy and the full marginal variance (200 scenarios per row for $G \leq 15$, 150 for $G=20$, 100 for $G=30$).*
>
> We report two quantities. Spearman's rank correlation $\rho$ measures cell-level agreement between the two variance vectors. Path-level agreement (PLA) is the fraction of scenarios in which $\arg\max$ PathMI selects the same direction under both proxies, at horizons $L=3$ and $L=5$. The time ratio $t_\text{full}/t_\text{diag}$ is the per-call wall-clock cost of direct inversion relative to the proxy. Spearman $\rho \geq 0.95$ across all grids confirms that cell-level ordering is preserved. PLA is $\geq 0.975$ at both horizons, so the exploration action selected by PathMI is the same under both proxies in at least $97.5\%$ of cases. The cost ratio grows from $\sim 60\times$ at $5\times 5$ to $\sim 1300\times$ at $30\times 30$, confirming that the proxy is required for operational-scale inference.
>
> We would be happy to include this analysis as a short appendix if the AE or reviewer feels it would strengthen the paper.
>
> ---
>
> **Comment 2.** *"For the PathMI objective, is there a way to derive this from an underlying information-theoretic quantity? A derivation of this (even in the appendix) could help give a more rigorous justification for the specific form considered here along with certain choices, such as $\xi\_c = 1/(1+n\_c)$."*
>
> **Response.** The reviewer is correct to suggest that the PathMI score may be derived from an underlying information-theoretic quantity. Specifically, it may be obtained from the expected information gain $I(\mathbf{u}; \mathbf{y}\_\mathcal{O} \mid \mathcal{D}\_t)$ under a sequence of assumptions, including, but not limited to, a Laplace–Gaussian posterior, a diagonal covariance approximation, and a small-information linearisation of the Gaussian mutual-information formula. We have not added a derivation to the manuscript as both information-theoretic criteria for nonmyopic active learning in Gaussian processes (Krause and Guestrin, 2007) and the use of predictive variance as the corresponding surrogate (MacKay, 1992; Riis et al., 2022) are established in the literature.
>
> ---
>
> **Summary of revisions.**
>
> 1. **Section 3.5.3** (diagonal Laplace approximation): one sentence added noting that direct comparison is tractable only at small grid scales, and that the proxy is adopted because it preserves relative uncertainty ordering.

---

### Decision · Action_Editor_YAhp · 2026-05-29

**Recommendation:** Reject

**Audience:**

Yes

**Audience Explanation:**

Those working on reinforcement learning, multi-agent coordination, informative path planning, and uncertainty-aware decision making may find the paper’s problem setting and hybrid LGCP/SAC framework of interest.

**Claims And Evidence:**

No

**Claims Explanation:**

The submission presents an interesting framework and includes several empirical comparisons and ablations, but the evidence does not adequately support the full scope of the paper’s claims. In particular, the manuscript makes broad claims about general multi-agent coordination under spatial uncertainty, while the evaluation is primarily limited to a simulated multi-UAV wireless service provisioning setting. Reviewers also raised unresolved concerns about the rigor and justification of key methodological components, including the PathMI objective, the diagonal uncertainty proxy, the reward design, and the generality of the framework beyond the tested setting. Although the authors’ response clarified some terminology and added discussion, the remaining evidence is not sufficiently accurate, convincing, and clear to substantiate the main claims at the level expected for publication.